# Study of the Photosynthesis Response during the Gradual Lack of Water for 14 *Olea europaea* L. subsp *europaea* Cultivars and Their Adaptation to Climate Change

**DOI:** 10.3390/plants12244136

**Published:** 2023-12-12

**Authors:** Genoveva Carmen Martos de la Fuente, Benjamín Viñegla, Elena Illana Rico, Ana Maria Fernández Ocaña

**Affiliations:** Departamento de Biología Animal, Biología Vegetal y Ecologia, Facultad de Ciencias Experimentales, Campus de Las Lagunillas s/n, Universidad de Jaén, 23071 Jaén, Spain; gmartos@ujaen.es (G.C.M.d.l.F.); bvinegla@ujaen.es (B.V.); eir00001@red.ujaen.es (E.I.R.)

**Keywords:** *Olea europaea*, drought stress, photosynthesis–light response, olive cultivars, cultivar characterisation

## Abstract

Understanding the tolerance of plants to drought and their gradual response to lack of water is a multifaceted challenge that requires a combination of scientific research and technological innovation. Selecting naturally drought-tolerant plants and knowing their response to photosynthesis in a wide range of water availability opens a door to making decisions about the suitability of different cultivars to be implanted in specific geographical areas, based on their tolerance to drought and light absorption capacity. In this work, photosynthesis–light curves were carried out using a LiCor LI-6800 IRGA device, applying increasing light intensities to plants of 14 olive cultivars, either under control conditions (no water stress) or subject to moderate and severe water deficits. The plants were grown in a culture chamber under controlled conditions for photoperiod, air humidity, temperature, and carbon dioxide concentration. For each cultivar, the electronic transference ratio (ETR) in response to light was also obtained. Different equations were used to fit experimental data allowing us to calculate, with a regression coefficient above 0.95, different photosynthetic parameters such as the maximum photosynthetic capacity, the photosynthetic efficiency, the number of electrons or the number of photons to assimilate a molecule of CO_2,_ and the effect of the lack of water on these parameters. This work represents the first contribution of the response to photosynthesis of many olive cultivars subjected to moderate and severe drought conditions. The parameters described, and the results provided, pave the road for subsequent work related to plant physiology and other areas of science and technology, and allow us to objectively compare the tolerance to water stress in these fourteen olive cultivars.

## 1. Introduction

Genetic resources are a global asset of inestimable value for the survival of future generations. The UN Convention on Biological Diversity (1992) [1] recognises that their conservation, which currently is being lost at a worrying rate, is of critical importance to humankind. In the context of climate change, one of the pressures accelerating biodiversity loss is habitat destruction (e.g., deforestation [2], urbanisation, and conversion of land for agriculture), leading to the loss of critical habitats for many species [3]. The other factors of pressure are overexploitation, pollution, and invasive exotic species [4]. A turn toward a green and sustainable economy is increasingly necessary. In the case of olive groves, the presence of monospecific crops established throughout the Mediterranean basin prevents autochthonous species from persisting over time, being relegated to their conservation in germplasm banks [5]. More than 1200 different cultivars have been described throughout the Mediterranean basin [6]. Additionally, there exists enormous biodiversity in the wild olive [7], a species relegated to the margins of olive groves and relict zones in the Mediterranean Basin. These genetic resources increase the possibility of olive breeding [8,9] in a new scenario of adaptation to climate change. This possibility is especially interesting for the study of some agronomic characters related to biotic and abiotic stresses. Specifically, and as an example, while there are no cultivated olive varieties that resist the wilt caused by *Verticillium dahliae*, some authors have studied the existence of wild germplasm capable of tolerating the presence of this fungus without producing disease in the plant [10,11].

In Spain, more than 250 olive cultivars have been reported [12,13]. The most abundant in the olive groves of the Iberian Penisula is Picual [14], typical of the fields of Jaén. Arbequina cultivar is one of the most abundant in Northeastern Spain [14], which, in recent years, has been widely used, together with Arbosana, Sikitita1, and Sikitita2 for crops grown in hedgerows and superintensive crops. Another new cultivar is Martina, obtained from crosses between other cultivars. Hojiblanca and Cornicabra are cultivars very common in the Málaga and Córdoba provinces [14], respectively. Manzanilla de Sevilla and Cornezuelo de Jaén are widely used for table olives [14]. Empeltre, cultivated in areas of Aragon, Navarra, and Castellón [14]. Other very common cultivars in other countries of the Mediterranean basin are Koroneiki, the most important cultivar in Greece, Frantoio, the main Italian variety, and Chemlali de Kabile, one of the most important varieties grown in Tunisia [14].

The improvement in the efficiency of photosynthesis for increasing crop yields is an issue of special attention between plant physiologists and breeders. In photosynthesis, there exists a balance between the energy used for photosystems II and I to obtain ATP and NADPH and the one spent to protect the photosynthetic machinery when the irradiances are very high [15]. The quantity and quality of light and the concentration of CO_2_ in the ambient have a direct effect on this process. At low irradiances, the amounts of energy produced in the light reactions are not sufficient and plants decrease and even stop their growth [16,17]. However, when the irradiances are very high, as occurs in the spring and summer of the Mediterranean Basin, another problem appears, since the plants need to use their photosystems efficiently to benefit from the abundant light that reaches them while avoiding excess irradiance, which produces damages in the structure of the photosynthetic apparatus [18]. Plants respond to water scarcity in different ways, with drought being a complex process that we still need to understand deeply [19]. The process of leaf anatomy development, i.e., the establishment of the polarity, xylem, and phloem formation, leaf blade expansion, shape morphogenesis, and the development of mesophyll (palisade and spongy tissues), have a great influence on the photosynthetic efficiency of plants [20]. If the process of formation of new tissues occurs in conditions of lack of water for the plant, growth slows down and existing leaves lose turgor and curve backward, decreasing the available light receiving area [21,22,23]. Thus, new leaves will be thinner, with fewer layers of palisade parenchyma, and lack of water can even cause forming fruits to fall or prevent them from fattening properly. When plants experience a lack of water, they close their stomata to conserve water [24] reducing the availability of carbon dioxide for photosynthesis. The production of oxygen free radicals interacts with the biochemical cell cycles. The chlorophyll content in leaves may decrease, causing damage to the photosynthetic machinery [25] and thus affecting the functionality of photosystems II and I. The enzymatic activity, synthesis, and structure of some proteins, such as RuBisCo and Phosphoenol pyruvate carboxylase, are affected, disrupting the overall efficiency of the photosynthetic process [16]. In a recent article, we have published evidence of a decrease in the photosynthetic capacity of several *Olea europaea* cultivars under water stress conditions measured in saturating irradiances [23]. However, this work goes further by measuring photosynthesis–light response curves in the three experimental conditions, that is, the gradual evolution of the response in photosynthesis to the lack of water in *Olea europaea* for different cultivars. In modern agriculture, farmers and agricultural practitioners need a deep understanding of plant traits, performance, and specific requirements of different cultivars to make informed decisions and optimise their agricultural practices.

In the studies carried out by Fernández et al. (2006), mature “Manzanilla” olive trees showed a clear limitation of photosynthesis due to the lack of water [26]. Other authors, such as Sofo et al. 2009, worked with the “Coratina” cultivar, measuring photosynthesis under drought conditions, also perceiving a gradual decrease in the plant’s assimilation capacity as the severity of the lack of water increased [27]. A drastic decrease in photosynthesis with photoinhibition was observed in young olive leaves subjected to continuous illumination compared to shaded leaves of the same cultivars [25]. According to Diaz-Espejo et al. (2007), diffusional limitation of photosynthesis, rather than light, determines the distribution of photosynthetic capacity in olive leaves under drought conditions [28]. Bacelar et al. (2009) tested the photosynthetic capacity of five olive cultivars subjected to drought [29]. According to their conclusions, Cobrançosa, Manzanilla, and Negrinha were better acclimated to drought conditions producing a high photosynthetic rate than Arbequina and Blanqueta, which appeared to show a conservative water-use strategy.

New perspectives to improve photosynthetic efficiency include a better display of leaves in crop canopies, avoiding the photorespiration process, especially in C3 species, genetic engineering of carboxylase enzymes to obtain a better affinity for CO_2_, and optimization of plants to maximise carbon gain with decreases in water use [30]. Few studies exist to date in this regard on *Olea europaea*. Aranda Barranco et al. (2020) [12,13], studied the capacity to sequester carbon in two Spanish olive groves whose management differed in the vegetation cover of each of them. In the first case, the field did not have such plant groundcovers compared to the second olive grove, which did. The experiment was carried out for a year, concluding that weed cover is responsible for the high carbon sequestration capacity of this conservation practice in olive groves. With the policies of adaptation to climate change, a crucial objective nowadays is the achievement of negative CO_2_ emissions to the atmosphere. Biotic sequestration can be a winning strategy for climate mitigation. Terrestrial ecosystems have the potential to sequester around 333 Gigatons of C by the end of this century [31,32]. It is necessary to implement the best management practices in agriculture, adapting olive groves to climate change in the Mediterranean basin, and their sustainability in olive oil production. The diversification of olive groves with a diverse range of cultivars can enhance resilience because of their varying tolerance to extreme temperatures, water stress, salinity, and diseases. By selecting cultivars, farmers minimise the risk of crop failure, provide better natural pest control and pollination, and improve overall ecosystem health. Planting groundcovers, creating a habitat for beneficial insects, and preserving native vegetation could help support a diverse and resilient ecosystem. In addition, the modification of carbon photoassimilation in crops through genetic engineering is a promising alternative, which can improve carbon sequestration in ecosystems [31].

Using genetic resources to solve problems such as those mentioned above, caused by climate change processes, is the best way to prevent them. For the design of new crop fields, both researchers and farmers, agricultural entrepreneurs, nursery workers, and owners of olive farms must have scientific information available on the behaviour of common varieties. This is one of the objectives of the regulations of the current European common agricultural policy [33].

The hypothesis of this work aims to obtain relevant information from the comparison of the behaviour of 14 olive cultivars subjected to water deficit during an interval of more than 30 days under water deprivation conditions. Curves have been measured in triplicate every seven days in the plants, in order to obtain the response capacity of each cultivar to a progressive water deficit. To objectively know the tolerance of each cultivar to drought, plants of the same age previously grown in the same type of soil (with the same field capacity) have been used, under fully controlled conditions, bringing them to the permanent wilting point. The information obtained from the results of this work will be a crucial starting point for many subsequent works related to agrivoltaics, the study of climate change, the choice of olive cultivars in the different latitudes and longitudes of the earth, and the suitability of each cultivar to the climate in which it is grown. Surprisingly, these cultivars produce very different responses under standard irrigation and even more so under moderate and severe drought conditions. Therefore, through an in-depth study of the photosynthetic efficiency and other gas exchange and fluorescence parameters, we have attempted to answer this working hypothesis.

## 2. Results and Discussion

The assimilation of CO_2_ in moderate and severe drought was studied in 14 different olive cultivars from all over the Mediterranean basin. The control plants (six samples of each cultivar) were grown in a growth chamber with water ad libitum. Another six replicates of each cultivar were grown in the same growth chamber but without any irrigation for at least 28 days for the sensitive cultivars, and up to 42 days without irrigation for the most tolerant cultivars. In the case of control plants, at least five response curves were performed per cultivar during the experiment. This behaviour represents the ability of each cultivar to respond to light intensity under optimal conditions, which were labelled as control conditions. To characterise the responses of these cultivars under waterless conditions, triplicate curves were obtained weekly for each cultivar. From the response curves obtained at 7, 14, and 21 days post-irrigation, the behaviour of the plant under MD conditions was determined. Likewise, from the curves obtained 21, 28, and 35 days after irrigation, the behaviour of each cultivar in the severe drought condition was determined. Each of the curves obtained was fitted to an ad hoc function, as explained in the section on materials and methods, to obtain photosynthetic parameters that describe the experimental data.

### 2.1. Curve Fitting

The curve exhibits several phases. At the beginning, from complete darkness to the light compensation point, there is a rapid increase in photosynthesis (P), due to the activation of photosynthetic processes as soon as light becomes available. As light intensity continues to increase beyond the light compensation point (quantity of light in which P is zero), the rate of photosynthesis increases, known as the “light-limited phase”. There comes a point at which an increase in light intensity does not lead to an increase in the value of P. This phase is called the “light-saturated phase” and is characterised by photosynthesis being limited by other factors, such as CO_2_ availability, temperature, or other biochemical processes.

The maximum photosynthetic rate (Pmax), the efficiency of photosystem II (ϕ_PSII), and the respiration rate in the dark (Rd) were determined in the three experimental conditions: control plants, MD, and SD, as indicated in materials and methods (Table 1). Additionally, other photobiological parameters such as theta (θ), the curvature of the curve between the linear increment phase and the saturation phase of photosynthesis in response to irradiance, and Jmax, the maximum rate of electron transport in saturating light (µmol electrons m^−2^·s^−1^), which influence plant photosynthetic activity, are studied in relation to environmental conditions. θ represents the fraction of absorbed photons that are used for photosynthesis. In other words, it indicates the proportion of light energy that is converted into chemical energy during photosynthesis. This model is a better fitting alternative to describing the relationship between photosynthesis and light compared to the Farquhar, von Caemmerer and Berry (FvCB) model [34].

### 2.2. Photosynthesis-Light Response and ETR-Light Curves of the 14 Olive Cultivars

Figure 1 represents the three real curves (C, MD, and SD) obtained for each cultivar and the theoretical curves that were mathematically adjusted to each of the real curves. In most cultivars, control samples present higher maximal photosynthesis (Pmax) and a higher saturation irradiance (Isat) than samples subjected to moderate drought (MD), and these, in turn, explain higher values of maximal photosynthesis and saturation irradiance than severe drought samples (SD). This behaviour confirms that plants with sufficient water availability, moderate temperature, and adequate CO_2_ concentration have a greater capacity to photosynthesise efficiently, their stomata are open for the correct gas uptake, and chlorophylls, photosystems, and chloroplast electron transport chains show good functionality. Figure 1 indicates that, although all the cultivars studied belong to the same species, *Olea europaea* subsp *europaea*, their behaviour is very different. It can be observed that cultivars such as Chemlali (Ch), Empeltre (E), and Cornicabra (C) produce high assimilation of CO_2_ (above 12 µmols of CO_2_ m^−2^·s^−1^). Comparatively, Manzanilla de Sevilla (MS) does not exceed 2.5 µmols.m^−2^·s^−1^ and Hojiblanca (H) does not assimilate more than 6 µmols.m^−2^·s^−1^ under control conditions. It is striking that some of the cultivars photosynthesise more efficiently under MD conditions than under control conditions. This is the case of MS, Frantoio (F), and Martina (M). Knowing the drought resistance of the olive tree, which is a typically Mediterranean species, it is not surprising that some olive cultivars may be more efficient when water availability is relatively reduced, as in the case of moderate drought conditions. This peculiarity is not directly related to drought tolerance: while K (for which almost no differences in assimilation between the control and MD conditions were found) is one of the most sensitive cultivars, Martina is one of the most tolerant. Boussadia et al. (2008) measured fluorescence at a saturation irradiance (Isat) of 1500 micromols.m^−2^·s^−1^ in two cultivars, one of which was Koroneiki [35]. The results of these authors on assimilation, stomatal conductance, and photorespiration were similar to those obtained by us. Assimilation gradually decreased from control plants to plants subjected to MD, but this decrease was much more drastic in the SD condition.

The behaviour of the cultivars follows several different patterns: there are some cultivars in which a small water deficit leads to a very sharp drop in photosynthesis. The most striking case occurs in Ch, which goes from being the most photosynthetically efficient plant under control conditions, assimilating more than 14 µmols of CO_2_ m^−2^·s^−1^, to producing almost no CO_2_ assimilation under MD and SD conditions. Although not to the extent of the Ch cultivar, the same pattern occurs in Arbequina (A), E, F, H, and Sikitita2 (S2). Boughalleb et al. (2011) studied the behaviour of young plants of two olive cultivars under drought, one of which was Chemlali. According to these authors, the absence of signs of drought sensitivity (leaf drop) under the water stress regime of this cultivar reinforces the idea that it can be grown in semi-arid regions and classified as a mild-risk cultivar to be planted in dry areas [36]. A very similar conclusion can be drawn from the data provided by Chemlali in this study. However, according to our results, this cultivar, despite having a very high Pmax in optimal growth conditions, shows a drastic decrease in its maximum photosynthesis when experiencing a slight lack of water. The plant, however, remains stable throughout the process of a gradual increase in lack of water. Therefore, in our opinion, Chemlali is quite resistant to drought cultivars, since it resisted without any irrigation until the end of the experiment without showing any apparent sign of serious damage to its leaves and stems. However, in the event of a lack of water, it is no longer photosynthetically efficient, and its assimilation rate drastically decreases. This is not the case with other cultivars, which show quite a capacity for assimilation even under very extreme conditions of lack of water. Other cultivars reduce their assimilation capacity gradually as the lack of water in the plant increases. This pattern of behaviour occurs in Arbosana (Ab), C, Cornezuelo de Jaén (CJ), K, M, MS, Picual (P), and Sikitita 1(S1). It should be noted that some cultivars, despite the extreme lack of water, continue to produce considerable CO_2_ assimilation in SD. This is the case of C, which produces a maximum photosynthesis of 6 µmols of CO_2_ m^−2^∙s^−1^ in SD or M, which produces 4 µmols of CO_2_ m^−2^∙s^−1^ under the same conditions. MS is the least photosynthetically efficient cultivar in control conditions, although the lack of water does not affect its assimilation capacity considerably in MD or SD.

**Figure 1 plants-12-04136-f001:**
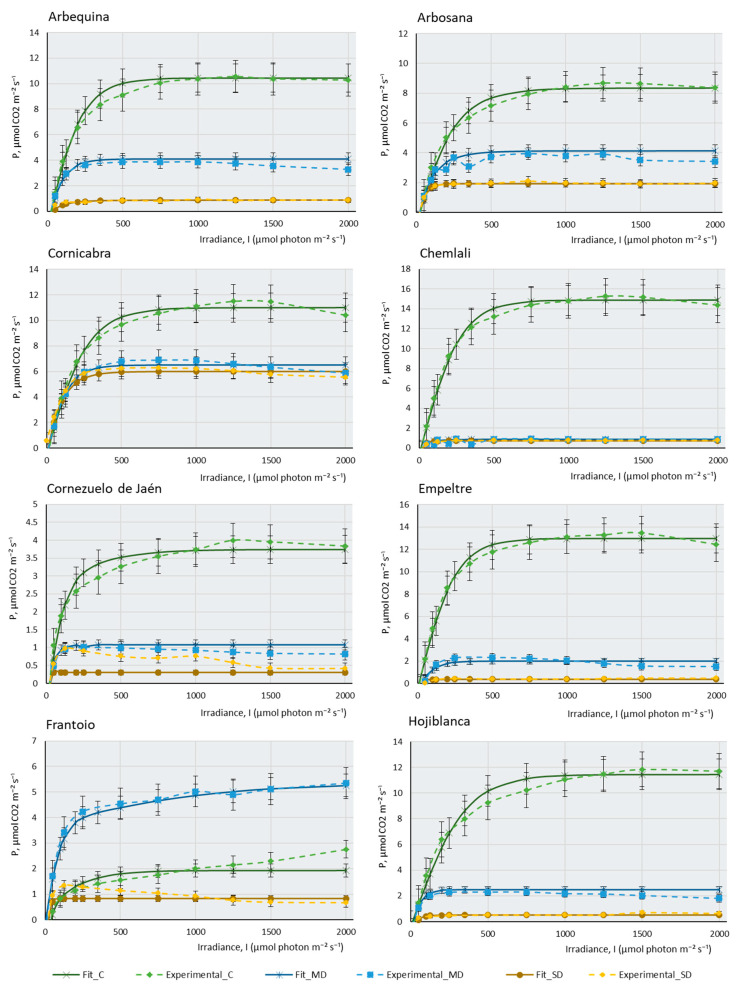
Experimental data of photosynthesis/light (P/l) curves (dotted lines) of the 14 olive cultivars studied, and their mathematical fitting under control and two water stress conditions. The fitting was carried out following the mathematical model described by [37]. The photosynthetic rate (P, µmol CO_2_ m^−2^·s^−1^) is shown as a function of the photosynthetic photon flux density (I, µmol photons m^−2^·s^−1^) for each cultivar under control, moderate drought (MD), and severe drought (SD) conditions. The data points represent the mean values, selecting only those where the mathematical relationship provided by the model is highly predictive and closely fits the observed real values in the P/I curve (R_2_ > 0.95). Error bars indicate the typical error considering the sample size of each experimental condition (nC = 4–6, nMD = 9, and nSD = 5–9). The curves display the photosynthetic response to different light levels (0–2000 µmol photons m^−2^·s^−1^) and irrigation conditions (C, MD, and SD). The photosynthetic response to light intensity varies among cultivars and drought conditions, suggesting differences in the adaptation of cultivars to water availability.

A similar article has recently been published on seedlings of *Prunus sargentii* and *Larix kaempherii* [22], two species widely spread throughout Asia, especially in Japan and Korea. These authors studied the photosynthesis–light response curves under drought conditions and, in parallel, they also studied the response of photosynthesis to different CO_2_ concentrations. Both species showed an increase in their photosynthesis rates under increasing incident irradiance. However, these authors also found a decrease in their maximum photosynthesis rate (Pmax) under drought due to the depletion of soil water content, showing a response similar to ours. Dias et al. (2018) studied the tolerance of three Portuguese olive cultivars by measuring chlorophyll fluorescence [38]. Cobrancosa was the cultivar most tolerant to lack of water. Faraloni et al. (2011) used a rapid technique for measuring chlorophyll fluorescence to in vitro determine the drought tolerance of 24 olive cultivars [39]. Leaves were obtained from plants grown in vitro from 24 olive cultivars that were subjected to environmental dehydration in a controlled culture chamber for 24 h and after this time chlorophyll fluorescence was measured. Of all the cultivars studied, only Frantoio coincides with the cultivars studied in this work although not in vivo. These authors concluded that measuring these fluorescence parameters in vitro is a good technique to determine the tolerance of a cultivar to drought. Sofo et al. (2009) studied some photosynthetic parameters in two olive tree cultivars, Coratina and Blancolillo, subjecting them to water deficit and different light intensities for 21 days [27]. These authors obtained results very similar to those presented in this work. Photosynthesis decreased drastically with a lack of water. As in our case, high light intensities promoted the inhibition of photosynthesis.

This work measured the electron transfer rate (ETR) that occurs along the transport chain during the luminous phase of photosynthesis in the thylakoids (Figure 2), that is, the number of electrons that pass through PSII mainly, and also through PSI, as a consequence of the excitation produced by light on photosynthetic pigments that are able to produce energy in the form of ATP and NADPH in the plant.

Under drought conditions, ETR can be affected more severely or not, depending on the cultivar’s adaptation and tolerance to water stress. Water deprivation significantly impacts photosynthesis and electron transport along PSII, resulting in a reduction of the maximum electron transport speed (Jmax). In this case, PSII is less efficient (as an adaptive strategy to prevent damage from photo-oxidation under high light intensities during drought), affecting the activity and stability of these complexes. There is a decrease in the saturation irradiance (Isat) that produces Jmax. Another consequence of water stress is a decrease in the curve slope, indicating a minor transfer of electrons with increasing light intensity and, finally, water deficit produces an increased susceptibility to photo-inhibition.

The ETR curves at different light intensities (ETR/l) were obtained for each olive cultivar (Figure 2). Under control conditions, with ad libitum irrigation of the samples, it can be observed that most cultivars produce a maximum ETR, except for some cultivars, such as F, K, and MS, in which the efficiency of electron transfer is higher under MD conditions, although under ad libitum irrigation conditions, M cultivar suffers a considerable decrease of assimilation (Figure 1) (proof that it is a very drought-tolerant variety); however, it is curious that the ETR value under control conditions is higher than in MD. Therefore, the amount of photosynthesis this plant produces does not depend so much on an increased electron transfer along the oxidation–reduction chain, but perhaps on enzymes related to the Calvin cycle, which may not be fully functional when the amount of water the plant receives is high. Figure 2 indicates that, in general terms, the photosystems do not seem to be affected severely by the lack of water in the plant. Even so, although this electron transition occurs in the plant in all experimental conditions, including SD conditions, it does not translate into effective photosynthesis. The minimum amount of ETR produced in SD (except for the E cultivar, where electron transfer seems to be very affected), occurs in Picual, with values of 10 µmols of electrons m^−2^∙s^−1^. As an example, we can see that, in the case of F, the affectation produced in the capacity of the photosystems to transfer electrons is very small, (Emax in SD > 25 µmols electrons m^−2^∙s^−1^); however, the assimilation of CO_2_ produced by this cultivar under MD conditions and, especially, in SD, is strongly affected by the lack of water. This observation suggests, once again, that it is not the macro-complexes associated with the thylakoids (PSII, PSI, Cit b6f) structures and their electron transfer that cause this sudden decrease in photosynthesis during drought stress. It should be noted that in C, CJ, and F, electron transfer is practically unaffected by drought, with very similar response curves in the three experimental conditions. In some cultivars (CH, H, and S2), there are barely noticeable differences between both drought conditions.

The purpose of adjusting these theoretical curves from the curves measured with the Li-Cor 6800 is to obtain and calculate parameters that will serve as indicators of the capacity of each cultivar to cope with the lack of water. These parameters are shown in Table 1, Table 2 and Table 3. From the adjustment of the curves, provided that this adjustment has a correlation coefficient with a value greater than 0.95, very interesting parameters were calculated in the three experimental conditions in order to discern objectively the tolerance of each cultivar to drought.

Small discrepancies can be observed between the real values of Pmax, represented in Figure 1 and Figure 2, and the theoretical values of Pmax that these curves show after mathematical adjustment (which represents the maximum theoretical photosynthesis rate under ideal conditions provided by the CurveExpert Basic software v 2.2.3, necessary in Equation (1) described in the Materials and Methods) [40]. This could be due to several reasons. Firstly, the equation being used to model the relationship between irradiance (I) and photosynthesis rate (P) may be a simplified representation of reality and may not capture all the complexities of the system. In some cases, a more complex model might be necessary to accurately describe the behaviour of photosynthesis. Secondly, the experimental data collected may have limitations or biases that affect the model’s ability to fit them. Experimental conditions, such as carbon dioxide (CO_2_) concentration, temperature, humidity, and other factors, can affect the ability of photosynthesis to reach its maximum rate under ideal conditions. Thirdly, photosynthesis shows a saturation response, which means that, as the irradiance increases, the rate of photosynthesis increases until it reaches a maximum value, after which the rate of photosynthesis no longer increases. This could explain why the calculated values of Theorical_P do not reach the maximum value of “P” in the range of irradiances. Lastly, respiration in darkness (Rd) is not being considered, which eventually reduces (depending on its value) the rate of photosynthesis in the graph.

In summary, although the Pmax value represents the theoretical maximum rate of photosynthesis under ideal conditions, in practice, several factors can limit the ability of plants to achieve this maximum rate under all irradiance conditions. Photosynthesis is a complex process influenced by multiple variables, and it is important to consider these limitations when interpreting the results and fitting the model.

Table 1 shows that the maximum photosynthesis (Pmax) progressively decreases in general with the lack of water and, in turn, the saturation irradiances (Isat50 and Isat99.9) decrease in a positive correlation with it. This decrease is logical since drought stress directly affects the availability of CO_2_ inside the cells of the leaf as a consequence of the stomatal closure induced by ABA. The efficiency of photosystem II (ф_PSII) also decreases, indicating that, with the lack of water, a greater number of photons are necessary to fix a molecule of CO_2_. This result completely coincides with that obtained by Sofo et al. (2009) in two different olive cultivars subjected to water stress [27]. The maximum electron transport speed (Jmax, or the ability to assimilate CO_2_) decreases in all cases, becoming almost zero for cultivars like F in SD. Rd and MS present strikingly high respiration in the dark compared to the rest of the cultivars. This parameter seems to remain constant in most cultivars for all experimental conditions, except for C and F, where it decreases with drought. The same occurs with the Fv/Fm values, which remain constant or decrease slightly with water deficit in all cultivars. In the work of Faraloni et al. (2011) carried out on leaves of olive cultivars grown in vitro and separated from the rest of the plant 24 h before measurement, the Fv/Fm values decreased considerably with the lack of water unlike what happens in our study [39]. We suggest that senescence reactions also take place in the experiment of these authors, and therefore there is an additional structural and functional deterioration in the leaves of the measured cultivars. Sofo et al. (2009) also measured Fv/Fm and Pmax in conditions of water deficit, obtaining the same decreasing trend as that mentioned by us in Coratina and Blancolillo cultivars [27].

Table 2 shows the values of some parameters obtained from the fitted curves of ETR response versus irradiance (Figure 2). Although the mathematical fit of these curves with the real curves is not as good as in the first case (photosynthesis–irradiance response), the calculation of the parameters indicated in Table 2 was obtained from the zones of the curve in which there is a fit with a value of R > 0.95 with the real curves. ETRmax is the maximum electron transfer rate in the thylakoids during luminous reactions. This parameter, as can be seen in Table 2, decreases with drought, which indicates the importance of water in the light phase of photosynthesis. The plant, by not having good functionality, decreases its capacity to receive light and transforms it into stable chemical energy. This occurs in all cultivars, except in F, K, and MS, where the maximum radius of electron transfer occurs under MD conditions. Alpha-ETR is the slope of the curve, that is, the efficiency of photosynthesis in relation to incident light intensity (measured in micromoles electrons/micromols photons). This efficiency is quite stable with the lack of water, which indicates that a practically constant number of electrons jump in the protein complexes of the chloroplast with a photon of light, regardless of the water deficit. Of course, the result produced on the ETRmax is different, since the efficiency of transferring those electrons from one macro-complex to another is affected by the lack of water. For example, in the case of the CH cultivar, although alphaETR (α) remains almost constant in the three experimental conditions, ETRmax goes from 88.9 under control conditions to 22.9 in MD and 26.1 in SD. The exception occurs in two cultivars, CJ and P, in which α decreases with drought. In the photosynthesis–light response curves carried out by Bhusal et al. (2020), these authors measured the maximum photosynthesis (Pmax) and the maximum electron transfer rates of PSII (ETRmax) in the same three experimental conditions as those carried out in this study [22]. Both parameters are directly correlated with the lack of water in the plants so their values gradually decrease from the control plants to those subjected to MD and decrease even more in SD in the two species studied (*Prunus sargentii* and *Larix kaempferi*). Our results are completely consistent with those from these authors for the measurement of both parameters.

ETRd is the electron transport rate when the plant is adapted to the dark and, therefore, all the macrocomplexes inserted in the thylakoids are completely reduced. Logically, this parameter has a very low value, since the plant is in darkness and does not receive luminous energy. ISAT_ETR50 refers to the point at which the electron transport rate reaches 50% of its maximum value in response to light intensity. Therefore, ISAT_ETR75 and ISAT_ETR99.9 are the irradiances at which 75% and 99.9% of electron transfer occurs, respectively, between the protein macro complexes involved in the red-ox chain of the light phase.

With respect to the data reflected in Table 3, it should be noted that ΔF/Fv′ represents the photosynthetic efficiency of the plant adapted to light. The value of this parameter decreases with the lack of water in general terms. There are some exceptions, such as M, where the maximum efficiency occurs in MD conditions, and S1, where its maximum efficiency occurs in SD. Both cultivars turn out to be very drought tolerant. Moreover, Fv/Fm is always greater than ΔF/Fm′, since the first parameter is measured when the plant is adapted to darkness, where all macrocomplexes are in a reduced state. NPQ (Non-Photochemical Quenching) is the capacity of photosystems to dissipate excess energy as heat. To this end, the number of carotenoids that are part of the antennas of the photosystems, which are responsible for obtaining this excess energy and dissipating it into the atmosphere in the form of heat, play a fundamental role. qP (Photochemical Quenching) represents the fraction of photosystems that are in the reduced state, i.e., available for the photochemical process, that is, the fraction of light that is, at those moments, used to produce photosynthesis, and qN (Non-Photochemical Quenching) represents the fraction of reaction centres of PSII that are in an oxidised state, or the quantity of dissipated light. The number of photons necessary to make an electron jump increases with drought; this parameter, above all, is greatly increased in SD conditions. The number of photons required to fix a CO_2_ molecule is slightly increased between the control samples and the samples subjected to MD. In some cultivars, even a moderate lack of water in the plant favours the CO_2_ fixation process.

Finally, the number of electrons that the plant needs to transfer along the redox chain to produce the fixation of a CO_2_ molecule is also increased with the lack of water.

## 3. Materials and Methods

### 3.1. Plant Material

Fourteen olive cultivars were used as plant material: Arbequina (A) Arbosana (Ab), chemlali (Ch) Cornicabra (C) Cornezuelo de Jaén (CJ), Empeltre €, Frantoio (F), Hojiblanca (H), Koroneiki (K), Manzanilla de Sevilla (MS), Martina (M), Picual (P), Sikitita1 (S1), and Sikitita2 (S2). These are predominant cultivars in the olive groves of the Mediterranean Basin, and they are used in the three culture systems of current olive groves: classical, intensive, and super-intensive production systems. There were 12 samples per cultivar of one year of age, from a height of one meter approximately, all of them certified by the World Olive Germplasm Bank of Córdoba (WOGBC). The cultivars used were Arbequina, Arbosana, Chemlali, Cornezuelo de Jaén, Cornicabra, Empeltre, Frantoio, Hojiblanca, Koroneiki, Manzanilla de Sevilla, Martina, Picual, Sikitita1, and Sikitita2. The experiment was carried out in a growth chamber Aralab, 12000 PHL-LED, with irrigation ad libitum for six “control” samples and the following controlled parameters: photoperiod light/dark 16 h/8 h; light intensity 1600 µmol m^−2^·s^−1^; temperature 21 °C; 400 ppm CO_2_ concentration in the chamber; and relative air humidity of 50%. The other six samples of each cultivar were used in the drought stress experiment. In this case, the plants stopped being watered for a minimum of 28 days in the case of the most sensitive cultivars and a maximum of 42 days for the most tolerant to water stress.

### 3.2. Curve Fitting

In order to obtain a full overview of photosynthetic performance in the different cultivars considered, we used different approaches for the fitting of the experimental data. Thus, firstly, we fitted the data using the equation described by [41] (Equation (1)), a hyperbolic tangent model:(1)$$P=P_{max}\cdot tanh\left(\phi_{PSII}\cdot \frac{I}{P_{max}}\right)-R_{d}$$
where P is the net assimilation rate (µmol CO_2_ m^−2^·s^−1^), Pmax is the maximum rate of net photosynthesis (µmol CO_2_ m^−2^·s^−1^), tanh is the hyperbolic tangent function, I is the incident irradiance (µmol photon m^−2^·s^−1^), $\phi_{PSII}$ is the initial slope of net photosynthesis versus irradiance (µmol CO_2_ µmol photon^−1^), and R_d_ is the respiration rate in darkness (µmol CO_2_ m^−2^·s^−1^).

In order to determine saturating irradiance (Isat), i.e., irradiance at which net assimilation rate was saturated, the assimilation rate versus irradiance data was also fitted according to the equation described by [42]
(2)$$I_{sat(n)}=arctanh\left(\frac{\frac{n}{100}\cdot \left(P_{max}-R_{d}\right)+R_{d}}{P_{max}}\right)\cdot \frac{P_{max}}{\phi_{PSII}}$$
where Pmax is the maximum net assimilation rate (µmol CO_2_ m^−2^·s^−1^), n is the fraction of Pmax for which I_sat(n)_ is calculated (dimensionless, [0 ≤ n ≤ 100]), I_sat(n)_ is the irradiance at which the n fraction of Pmax is produced (µmol photon m^−2^·s^−1^), R_d_ is the respiration rate in darkness (µmol CO_2_ m^−2^·s^−1^), and $\phi_{PSII}$ is the initial slope of net photosynthesis versus irradiance (µmol CO_2_ µmol photon^−1^). Thus, this equation allowed us to calculate I_sat(100)_, I_sat(75)_, and I_sat(50)_, the irradiances at which 100% (typically known as saturating irradiance), 75% and 50% of Pmax were achieved, respectively.

Electron Transfer Rate was calculated through the following equation:(3)$$ETR=AQ\cdot FII\cdot AF/Fm^{\prime }$$
where ETR is the electron transport rate (µmol e^-^ m^−2^·s^−1^), AQ is the absorbed quanta (µmol photon m^−2^·s^−1^), calculated as the product of incident quanta by absorptance, which is considered to be 0.5, and AF/Fm′ is the effective quantum yield of PS II.

The equation proposed by [41] was also used to fit the data from the fluorescence vs. irradiance curves (Equation (2), modified from Equation (1)):(4)$$ETR={ETR}_{max}\cdot tanh\left(\alpha_{ETR}\cdot \frac{I}{{ETR}_{max}}\right)-{ETR}_{d}$$
where ETR is the electron transport rate (see Equation (3), defined in the above equation) (µmol e^−^ m^−2^·s^−1^), ETRmax is the maximum electron transport rate (µmol e^−^ m^−2^·s^−1^), tanh is the hyperbolic tangent function, I is the incident irradiance (µmol photon m^−2^·s^−1^), $\alpha_{ETR}$ is the initial slope of ETR vs irradiance (µmol e^−^ µmol photon^−1^) and ETR_d_ is the electron transport rate in darkness (µmol e^−^ m^−2^·s^−1^).

Additionally, experimental data of net photosynthesis versus irradiance were fitted using Equation (5) described by [43]:(5)$$P=\frac{\left(I+J_{max}\right)-\sqrt{{\left(I+J_{max}\right)}^{2}-4\cdot \theta \cdot I\cdot J_{max}}}{2}\cdot \theta$$
where P is the net assimilation rate (µmol CO_2_ m^−2^·s^−1^), I is the incident irradiance (µmol photon m^−2^·s^−1^), J_max_ is the maximum rate of electron transport (µmol e^−^ m^−2^·s^−1^) needed to supply energy for carboxylation and reduction reactions, which are subsequently catalysed by RuBisCO in the C assimilation stage (Calvin-Benson cycle), and θ is the curvature factor (convexity (dimensionless), [0 < θ < 1]).

Finally, several parameters were calculated from those obtained from the curve fitting.

### 3.3. Photosynthesis and Fluorescence versus Light Response Curves

All gas exchange and fluorescence measurements were recorded on fully expanded adult leaves in plants from each cultivar. The photosynthesis response (net photosynthetic rate, P, µmol CO_2_ m^−2^·s^−1^) to different irradiance (µmol photon m^−2^·s^−1^) was measured on 5 leaves (either control or moderate and severe drought samples), using a LI-6800 Portable Photosynthesis System with a transparent leaf chamber. The chamber was installed in a direction vertically upward to the soil surface using a level so as to not affect the other leaves. The LI-6800 was equipped with a 6800-01 fluorometer, to determine fluorescence parameters. Here is a brief explanation of each parameter: Fv/Fm is an estimate of the maximum quantum efficiency of PSII reaction centres. This ratio is calculated from two parameters: Fo and Fm. Fo is the fluorescence level of a dark-adapted plant with all PSII primary acceptors ‘open’ (QA fully oxidised). Fm is the maximal fluorescence level achieved upon application of a saturating flash of light, such that all primary acceptors ‘close’ Quinone QA fully reduced. Variable fluorescence, Fv, is the difference between Fo and Fm. Pmax (also known as Psat or PN, µmol CO_2_ m^−2^·s^−1^) refers to the maximum rate of net CO_2_ assimilation per unit leaf area. It represents the highest achievable photosynthesis rate of a leaf under optimal conditions, such as high radiation, suitable temperature, and no limitations from other factors. R_d_ (also known as R_dark_ or R, µmol CO_2_ m^−2^·s^−1^) is the rate of dark respiration. It denotes the amount of CO_2_ released by plant respiration in the absence of light. R_d_ is a constant rate and is a crucial component in calculating the net photosynthesis rate under illuminated conditions. Photosynthetic efficiency, alpha (α, µmol CO_2_ µmol photon^−1^), represents the photosynthetic efficiency. It indicates the plant’s ability to convert absorbed radiation into photosynthesis. Alpha corresponds to the initial slope (linear response) of the photosynthesis response curve concerning radiation. Theta (θ, dimensionless) is the curvature of the curve response in the transition from the linear part to the saturation part, i.e., the rate at which photosynthesis saturates at high irradiances. This parameter is linked to photoinhibition, which is the reduction in photosynthesis rate at high light intensities. Jmax (µmol e^−^ m^−2^·s^−1^) stands for the light-saturated potential maximum rate of electron transport through photosystem II (PSII) measured from the evolution of CO_2_ assimilation. It represents the plant’s utmost capacity to utilise light in the photosynthesis process and is a critical limiting factor for photosynthesis at high light intensities.

These parameters are crucial for understanding and modelling plant responses to light and photosynthesis under diverse environmental conditions. Their estimation and comprehension are fundamental for studies in plant physiology and ecology, and they have practical applications in areas such as agronomy and conservation biology.

Before recording the photosynthetic and fluorescence measurements, five randomly selected plants were adapted to darkness overnight. Then, saturated pulsed light (3000 μmol m^−2^·s^−1^, 300 ms duration) was applied to determine maximum quantum yield efficiency (Fv/Fm), according to Equation (6) [44]:(6)$$\frac{F_{v}}{F_{m}}=\frac{F_{m}-F_{o}}{F_{m}}$$
where Fv is the variable fluorescence (dimensionless), Fm is the darkened maximum fluorescence (dimensionless), and Fo is the darkened minimum fluorescence (dimensionless). This measurement allowed adjusting parameters in the fluorometer system to properly acquire fluorescence signals.

Light response data were gathered using the “Autolog-Light response” auto-programme in the LI-6800. Full light response curves on trees from each olive cultivar and control/drought treatments were measured using 10 different incident irradiance levels (2000, 1500, 1000, 750, 500, 400, 200, 100, 50, 0 µmol photon m^−2^·s^−1^). All the plants in each combination were selected randomly. During the measurements, the following parameters were controlled: CO_2_ concentration at 400 ppm, relative humidity at 40%, ambient temperature at 21 °C, airflow at 500 mmol s^−1^, and fan speed at 10,000 rpm. The auto-programmes were set to run for 60 to 180 s at a given light level before moving to the next light level in the auto-programme. The responses of the 14 cultivars to photosynthesis subjected to drought stress and in the control treatment were measured once a week in triplicate to obtain nine response curves to light for the analysis of the moderate drought interval (7, 14, and 21 days post irrigation) and nine curves for the analysis of the severe drought interval (curves taken at 21, 28, and 35 days post irrigation) vs the control plants.

In each light response curve, the following parameters were estimated from the gas exchange: P, net assimilation rate (µmol m^−2^·s^−1^), gsw, stomatal conductance to water vapour (mol H_2_O m^−2^·s^−1^), E, and transpiration rate (mol H_2_O m ^−2^·s^−1^).

Using the fluorometer during the light response curves the following parameters were also determined: Fv/Fm, maximum quantum yield efficiency (as defined above, dimensionless, determined in darkness), ΔF/Fm′, effective quantum yield efficiency in the light-adapted plant (dimensionless, defined as in equation), qP, photochemical quenching (dimensionless, defined as in Equation (7)), and qN, non-photochemical quenching (dimensionless, defined as in Equation (7)).
(7)$$\frac{\Delta F}{F_{m^{\prime }}}\left(effective\;quantum\;yield\right)=\frac{F_{m^{\prime }}-F_{s}}{F_{m^{\prime }}}$$

Fm′ is the maximum fluorescence emission under steady-state illumination (dimensionless, analogous to Fm), and Fs is the minimum fluorescence emission under steady-state illumination (dimensionless, analogous to Fo).
(8)$$qP(photochemical\;quenching)=\frac{F_{m^{\prime }}-F_{s}}{F_{m^{\prime }}-F_{o^{\prime }}}$$
where Fo′ is the minimum fluorescence emission in a light-adapted sample after a brief dark period, and Fm′ and Fs as described above.
(9)$$qN(non\text{-}photochemical\;quenching)=\frac{F_{m^{\prime }}-F_{s}}{F_{m^{\prime }}-F_{o^{\prime }}}$$
with all the parameters as described above.

Although values of E, ΔF/Fm′, qP, and qN were determined in each curve for all the incident irradiances, only those measured at the saturating irradiance (I_sat(100)_) were considered.

Finally, electron transport rate curves (ETR) were considered. ETR was calculated according to Equation (10):(10)$$ETR(electron\;transport\;rate)=AQ\cdot FII\cdot \Delta F/Fm^{\prime }$$
where ETR is the electron transport rate (µmol e^−^ m^−2^·s^−1^), AQ is the absorbed quanta (µmol photon m^−2^·s^−1^), calculated as the product of incident quanta by absorptance, FII is the fraction of AQ directed to Photosystem II, and AF/Fm′ is the effective quantum yield of PS II. Absorptance was not measured but considered to be 0.84, the average ratio of light absorbed by leaves in higher plants, and FII was considered to be 0.5, as the average ratio of PSII to PSI reaction centres [45].

### 3.4. Statistical Analyses

Once all the study parameters were collected, an analysis of distribution and homogeneity of variances was performed as a function of the fixed factors: experimental condition and cultivar type. The study of normality and homogeneity of variances was based on the p-value result obtained from the Shapiro–Wilk test and Levene test, respectively. For *p*-values > 0.05 in both tests, it is considered that the data approximate a normal and homogeneous distribution.

For those parameters that showed a normal distribution and homogeneity in their variances, a parametric analysis was conducted, in this case, a factorial analysis of variance, to identify if there was interaction between the fixed factor “condition” and the fixed factor “cultivar” for each study parameter. Parameters showing positive interactions underwent a subsequent post hoc analysis to identify significant changes between experimental conditions for the same cultivar or between cultivars for a specific experimental condition.

For parameters that followed a non-normal distribution or heterogeneity in their variances, multiple transformations, such as arccosine, logarithmic, exponential, inverse, etc., were performed until a normal and homogeneous distribution was achieved. In cases where data transformation was not possible, a non-parametric analysis was conducted. The Kruskal–Wallis test was chosen as a non-parametric version of one-way ANOVA, as the data came from independent samples, but the assumptions of normality or homogeneity of variances were not met.

In both analyses, only significant changes with *p*-value < 0.05 were considered. The Tukey test (*p* ≤ 0.05) was applied for comparisons between groups in parametric analyses, and the Dunn’s multiple comparison test (an extension of the Mann–Whitney test) allowed comparisons between pairs of groups after the Kruskal–Wallis test (non-parametric analysis).

These statistical analyses were conducted using RStudio Version: 2023.06.1 + 524 (requires R 3.3.0+, 2023 Posit Software, PBC formerly RStudio, PBC) along with the statistical software Statgraphics Centurion 19 Version 19.1.3 (Statgraphics Technologies, Inc., P.O. Box 134, The Plains, Virginia 20198 1982-2020).

## 4. Conclusions

Having exhaustive data on the responses of each cultivar to a moderate and severe water deficit helps determine the capacity and efficiency of the olive trees to maximise their productivity, optimizing crop yields according to the geographical latitude and longitude at which they are grown. It is an essential starting point to develop future work in the field of Agronomy and Plant Production. This work represents a reference for professionals dedicated to olive groves since it demonstrates that the CO_2_ assimilation process in olive trees shows important variations depending on the cultivar and its adaptations to the environment. This knowledge will also provide valuable information to biotechnologists and genetic breeders about the adaptive capacity of plants to specific environments and will allow the selection of varieties more efficiently. Taking advantage of the available light in different environments will lead farmers and owners to a better and more sustainable design of agricultural systems with optimal use of resources, including sunlight. From an ecological point of view, understanding how photosynthesis varies throughout the day at different light intensities helps predict how these changes can affect the structure of the plant community and the dynamics of ecosystems in general. Efficient water management in agriculture is also relevant to develop more effective and sustainable irrigation strategies with more tolerant cultivars, as well as to make informed decisions about water use in different contexts.

Finally, these light curves are a tool without which it would not be possible to work in the new disciplines that prioritise obtaining green energy from sunlight, respecting already existing agricultural systems as occurs in Agrivoltaics [46].

According to the data obtained for the 14 cultivars in the two experimental drought conditions, we can make a classification based on their ability to photosynthesise at optimal light intensities and according to the tolerance expressed by each of them against water deficit.

Table 4 offers the possibility of choosing the most suitable cultivar to implement according to these two observed variables.

## Figures and Tables

**Figure 2 plants-12-04136-f002:**
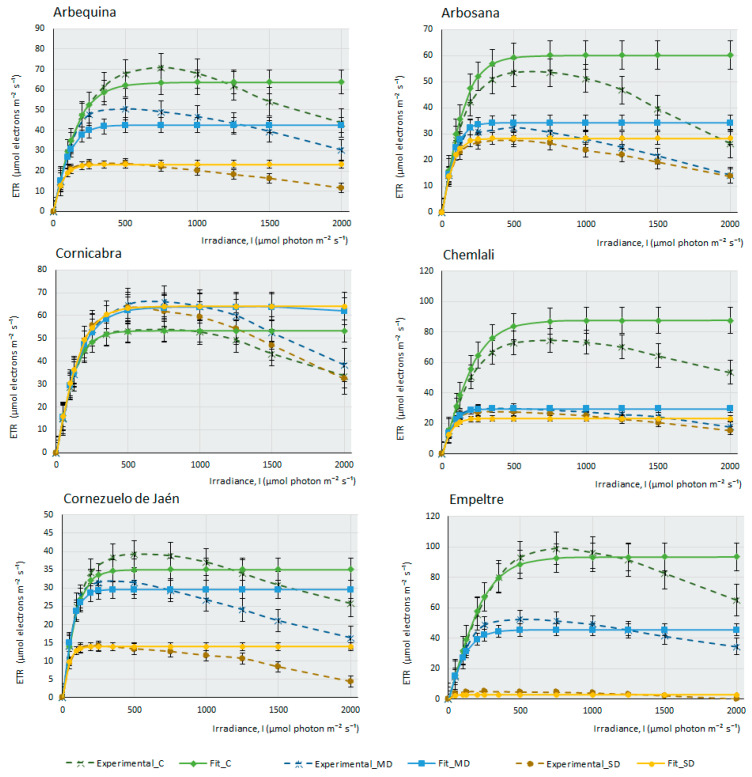
Fitting of ETR/I curves for 14 olive cultivars under different water stress conditions following the mathematical model described by [37]. The electron transport rate (ETR, µmol electrons m^−2^·s^−1^) is shown as a function of photosynthetic photon flux density (I, µmol photon m^−2^·s^−1^) for each cultivar under control, moderate drought (MD), and severe drought (SD) conditions. The data points represent the mean values, selecting only those where the mathematical relationship provided by the model is highly predictive and closely fits the observed values on the curve (R^2^ > 0.95). Error bars indicate the typical error considering the sample size of each experimental condition (nC = 4–6, nMD = 9, and nSD = 5–9). The curves depict the photosynthesis response at different light levels (0–2000 µmol photon m^−2^·s^−1^) and watering conditions (C, MD, and SD).

**Table 1 plants-12-04136-t001:** Mean values and standard deviation of the parameters obtained from the hyperbolic tangential model fitted to the ETR versus irradiance curve. In each parameter, different lowercase letters indicate significant differences among treatments (“Control”, “MD”, and “SD”) for the same cultivar. Different capital letters indicate significant differences among cultivars for the “Control” treatment, different capital letters with an apostrophe indicate significant differences among cultivars for the “MD” treatment, and different capital letters with two apostrophes indicate significant differences among cultivars for the “SD” treatment. Pmax maximum rate of net photosynthesis. φ_PSII quantum efficiency of photosystem II. Rd respiration in the dark. ISAT_A_50_ quantity of light at which the CO_2_ assimilation rate reaches 50% of its maximum value; ISAT_A_99.9_ would be 99.9% of photosynthesis or Pmax. Jmax maximum rate of electron transport; Theta is the curvature factor of the response curve of photosynthesis to radiation (dimensionless parameter), Fv/Fm relationship between the fluorescence variability Fv, t (Fm-Fo) with respect to the maximum fluorescence Fm.

Cultivar	Condition	Amax	φ_PSII	Rd	ISAT_A50	ISAT_A99.9	Jmax	Tetha	Fv/Fm
Name	Symbol	µmol CO₂ m^−^²·s^−^¹	µmol CO₂/µmol photons	µmol CO₂ m^−^²·s^−^¹	µmol photons m^−^²·s^−^¹	µmol photons m^−^²·s^−^¹	µmol CO₂ m^−^²·s^−^¹	-	-
AE	Control	11.15 ^a^ ± 1.72 ^ABC^	0.0497 ^a^ ± 0.0035 ^AB^	0.91 ^a^ ± 0.05 ^A^	145.0 ^a^ ± 7.3 ^A^	925.1 ^a^ ± 46.3 ^A^	205.5 ^a^ ± 31.08 ^B^	0.246 ^a^ ± 0.012 ^B^	0.827 ^a^ ± 0.041 ^A^
	MD	4.87 ^b^ ± 1.54 ^BCD’^	0.0464 ^a^ ± 0.0121 ^AB’^	1.16 ^a^ ± 0.06 ^AB’^	80.5 ^b^ ± 5.8 ^A’^	425.7 ^b^ ± 27.9 ^A’^	86.9 ^a^ ± 38.3 ^B’^	0.213 ^a^ ± 0.01 ^A’^	0.830 ^a^ ± 0.041 ^A’^
	SD	1.55 ^c^ ± 0.40 ^C’’^	0.0340 ^a^ ± 0.0166 ^C’’^	0.96 ^a^ ± 0.08 ^A’’^	66.7 ^b^ ± 11.0 ^A’’^	289.8 ^b^ ± 51.6 ^A’’^	27.7 ^a^ ± 6.9 ^A’’^	0.350 ^a^ ± 0.055 ^AB’’^	0.801 ^a^ ± 0.030 ^A’’^
Ab	Control	9.16 ^a^ ± 2.01 ^BC^	0.0332 ^a^ ± 0.0087 ^CD^	0.80 ^a^ ± 0.04 ^A^	114.3 ^a^ ± 5.7 ^A^	888.9 ^a^ ± 44.5 ^A^	222.4 ^a^ ± 46.3 ^B^	0.189 ^a^ ± 0.009 ^B^	0.829 ^a^ ± 0.041 ^A^
	MD	3.87 ^b^ ± 0.95 ^CD’^	0.0452 ^a^ ± 0.0151 ^ABCD’^	0.85 ^a^ ± 0.07 ^AB’^	72.0 ^a^ ± 6.4 ^A’^	415.7 ^a^ ± 42.3 ^A’^	75.7 ^a^ ± 37.6 ^B’^	0.146 ^a^ ± 0.008 ^A’^	0.775 ^a^ ± 0.038 ^A’^
	SD	2.60 ^c^ ± 0.68 ^BC’’^	0.0454 ^a^ ± 0.0076 ^C’’^	0.89 ^a^ ± 0.04 ^A’’^	96.8 ^a^ ± 20.5 ^A’’^	459.0 ^a^ ± 84.2 ^A’’^	47.9 ^a^ ± 18.5 ^A’’^	0.192 ^a^ ± 0.005 ^B’’^	0.751 ^a^ ± 0.036 ^A’’^
C	Control	10.92 ^a^ ± 2.09 ^ABC^	0.0438 ^a^ ± 0.0062 ^ABC^	0.77 ^a^ ± 0.04 ^A^	143.8 ^a^ ± 7.2 ^A^	921.9 ^a^ ± 46.1 ^A^	193.9 ^a^ ± 44.4 ^B^	0.118 ^a^ ± 0.006 ^B^	0.836 ^a^ ± 0.042 ^A^
	MD	6.92 ^b^ ± 1.88 ^AB’^	0.0512 ^a^ ± 0.0114 ^ABCD’^	0.59 ^a^ ± 0.07 ^B’^	95.2 ^a^ ± 36.4 ^A’^	611.3 ^a^ ± 255.7 ^A’^	86.7 ^b^ ± 15.6 ^B’^	0.154 ^a^ ± 0.285 ^A’^	0.806 ^a^ ± 0.039 ^A’^
	SD	6.45 ^b^ ± 1.98 ^A’’^	0.0436 ^a^ ± 0.0066 ^C’’^	0.11 ^a^ ± 0.01 ^A’’^	86.8 ^a^ ± 8.5 ^A’’^	611.1 ^a^ ± 89.6 ^A’’^	81.5 ^b^ ± 26.5 ^A’’^	0.348 ^a^ ± 0.085 ^AB’’^	0.818 ^a^ ± 0.039 ^A’’^
Ch	Control	14.82 ^a^ ± 1.72 ^A^	0.0562 ^a^ ± 0.0023 ^A^	0.58 ^a^ ± 0.03 ^A^	180.8 ^a^ ± 9.0 ^A^	1203.0 ^a^ ± 60.2 ^A^	200.5 ^a^ ± 37.9 ^B^	0.266 ^a^ ± 0.013 ^B^	0.804 ^a^ ± 0.040 ^A^
	MD	1.50 ^b^ ± 0.49 ^D’^	0.0336 ^a^ ± 0.0074 ^DCD’^	0.71 ^a^ ± 0.02 ^B’^	54.5 ^b^ ± 6.4 ^A’^	273.7 ^b^ ± 37.4 ^A’^	56.9 ^b^ ± 26.1 ^B’^	0.170 ^a^ ± 0.012 ^A’^	0.701 ^b^ ± 0.038 ^A’^
	SD	1.43 ^b^ ± 0.17 ^C’’^	0.0301 ^a^ ± 0.0081 ^D’’^	0.72 ^a^ ± 0.02 ^A’’^	32.9 ^b^ ± 2.1 ^A’’^	148.4 ^b^ ± 10.8 ^A’’^	42.2 ^b^ ± 11.3 ^A’’^	0.113 ^a^ ± 0.012 ^B’’^	0.723 ^ab^ ± 0.031 ^A’’^
CJ	Control	3.69 ^a^ ± 1.83 ^DE^	0.0400 ^a^ ± 0.0066 ^ABC^	0.61 ^a^ ± 0.04 ^A^	161.6 ^a^ ± 11.7 ^A^	810.9 ^a^ ± 57.5 ^A^	90.1 ^a^ ± 11.0 ^B^	0.059 ^a^ ± 0.009 ^B^	0.800 ^a^ ± 0.037 ^A^
	MD	1.80 ^b^ ± 0.45 ^D’^	0.0469 ^b^ ± 0.0072 ^CD’^	0.73 ^a^ ± 0.06 ^B’^	51.9 ^b^ ± 4.1 ^A’^	273.2 ^ab^ ± 27.7 ^A’^	63.1 ^a^ ± 19.9 ^B’^	0.340 ^a^ ± 0.034 ^A’^	0.781 ^a^ ± 0.706 ^A’^
	SD	1.05 ^b^ ± 0.08 ^C’’^	0.0451 ^b^ ± 0.0048 ^BC’’^	0.79 ^a^ ± 0.04 ^A’’^	6.4 ^b^ ± 1.4 ^A’’^	33.1 ^b^ ± 7.2 ^A’’^	57.2 ^a^ ± 6.4 ^A’’^	0.179 ^a^ ± 0.037 ^B’’^	0.648 ^a^ ± 0.028 ^A’’^
E	Control	12.99 ^a^ ± 2.40 ^AB^	0.0510 ^a^ ± 0.0067 ^AB^	0.56 ^a^ ± 0.03 ^A^	126.1 ^a^ ± 5.2 ^A^	825.0 ^a^ ± 33.1 ^A^	181.3 ^a^ ± 28.3 ^B^	0.268 ^a^ ± 0.013 ^AB^	0.826 ^a^ ± 0.000 ^A^
	MD	2.40 ^b^ ± 1.66 ^D’^	0.0613 ^a^ ± 0.0067 ^A’^	1.10 ^a^ ± 0.08 ^AB’^	56.5 ^ab^ ± 3.7 ^A’^	319.0 ^ab^ ± 22.9 ^A’^	132.5 ^a^ ± 51.9 ^B’^	-0.025 ^a^ ± 0.010 ^A’^	0.754 ^a^ ± 0.029 ^A’^
	SD	1.37 ^b^ ± 0.34 ^C’’^	0.0680 ^a^ ± 0.0070 ^A’’^	0.89 ^a^ ± 0.04 ^A’’^	23.1 ^b^ ± 3.2 ^A’’^	90.4 ^b^ ± 12.5 ^A’’^	82.6 ^a^ ± 2.8 ^A’’^	0.394 ^a^ ± 0.03 ^AB’’^	0.738 ^a^ ± 0.030 ^A’’^
F	Control	2.15 ^a^ ± 0.81 ^E^	0.0479 ^a^ ± 0.0103 ^ABC^	0.97 ^a^ ± 0.06 ^A^	180.7 ^a^ ± 7.7 ^A^	553.9 ^a^ ± 51.1 ^A^	232.9 ^a^ ± 46.3 ^A^	0.060 ^a^ ± 0.006 ^B^	0.820 ^a^ ± 0.000 ^A^
	MD	5.22 ^b^ ± 1.00 ^ABC’^	0.0540 ^a^ ± 0.0103 ^ABC’^	0.32 ^a^ ± 0.04 ^B’^	136.6 ^a^ ± 24.4 ^A’^	951.7 ^a^ ± 187.5 ^A’^	69.5 ^b^ ± 16.2 ^AB’^	0.398 ^a^ ± 0.049 ^A’^	0.781 ^a^ ± 0.000 ^A’^
	SD	1.69 ^a^ ± 0.48 ^BC’’^	0.0639 ^a^ ± 0.0053 ^A’’^	0.59 ^a^ ± 0.05 ^A’’^	100.7 ^a^ ± 2.0 ^A’’^	695.2 ^a^ ± 178.6 ^A’’^	1.2 ^b^ ± 0.6 ^A’’^	0.603 ^a^ ± 0.040 ^AB’’^	0.714 ^a^ ± 0.018 ^A’’^
H	Control	11.35 ^a^ ± 1.66 ^ABC^	0.0354 ^a^ ± 0.0040 ^BCD^	0.58 ^a^ ± 0.03 ^A^	65.9 ^a^ ± 5.8 ^A^	405.9 ^a^ ± 40.6 ^A^	260.4 ^a^ ± 36.3 ^B^	-0.019 ^a^ ± 0.001 ^B^	0.808 ^a^ ± 0.039 ^A^
	MD	2.74 ^b^ ± 0.81 ^D’^	0.0507 ^a^ ± 0.0134 ^BCD’^	0.67 ^a^ ± 0.05 ^AB’^	59.8 ^a^ ± 6.9 ^A’^	348.7 ^a^ ± 50.5 ^A’^	51.5 ^a^ ± 24.8 ^B’^	0.363 ^a^ ± 0.033 ^A’^	0.804 ^a^ ± 0.037 ^A’^
	SD	1.22 ^b^ ± 0.33 ^C’’^	0.0388 ^a^ ± 0.0099 ^CD’’^	0.68 ^a^ ± 0.05 ^A’’^	50.5 ^a^ ± 3.4 ^A’’^	253.1 ^a^ ± 30.6 ^A’’^	35.7 ^a^ ± 9.7 ^A’’^	-0.177 ^a^ ± 0.076 ^AB’’^	0.804 ^a^ ± 0.037 ^A’’^
K	Control	7.57 ^a^ ± 2.00 ^CD^	0.0173 ^a^ ± 0.0061 ^D^	0.55 ^a^ ± 0.06 ^A^	212.4 ^a^ ± 10.5 ^A^	1294.8 ^a^ ± 71.8 ^A^	345.3 ^a^ ± 55.7 ^B^	0.093 ^a^ ± 0.008 ^B^	0.799 ^a^ ± 0.039 ^A^
	MD	6.82 ^a^ ± 1.25 ^ABCD’^	0.0449 ^ab^ ± 0.0105 ^ABCD’^	0.63 ^a^ ± 0.03 ^AB’^	79.9 ^b^ ± 5.6 ^A’^	509.3 ^b^ ± 39.4 ^A’^	89.6 ^b^ ± 39.4 ^B’^	0.151 ^a^ ± 0.016 ^A’^	0.696 ^b^ ± 0.038 ^A’^
	SD	1.05 ^b^ ± 0.09 ^C’’^	0.0481 ^b^ ± 0.0174 ^C’’^	0.78 ^a^ ± 0.02 ^A’’^	27.6 ^b^ ± 2.3 ^A’’^	157.9 ^b^ ± 13.5 ^A’’^	30.2 ^b^ ± 9.2 ^A’’^	0.295 ^a^ ± 0.029 ^AB’’^	0.680 ^b^ ± 0.039 ^A’’^
M	Control	2.96 ^a^ ± 0.97 ^DE^	0.0471 ^a^ ± 0.0050 ^ABC^	0.65 ^a^ ± 0.03 ^A^	118.4 ^a^ ± 7.6 ^A^	702.9 ^a^ ± 50.7 ^A^	47.8 ^a^ ± 17.9 ^B^	1.614 ^a^ ± 0.159 ^A^	0.803 ^a^ ± 0.040 ^A^
	MD	9.55 ^b^ ± 2.20 ^A’^	0.0460 ^a^ ± 0.077 ^BCD’^	0.66 ^a^ ± 0.02 ^B’^	112.9 ^a^ ± 3.9 ^A’^	699.7 ^a^ ± 42.5 ^A’^	127.6 ^b^ ± 36.5 ^B’^	-0.012 ^b^ ± 0.006 ^A’^	0.772 ^a^ ± 0.002 ^A’^
	SD	4.43 ^a^ ± 1.29 ^AB’’^	0.0385 ^a^ ± 0.0081 ^C’’^	0.94 ^a^ ± 0.05 ^A’’^	78.1 ^a^ ± 4.4 ^A’’^	414.5 ^a^ ± 24.0 ^A’’^	95.9 ^ab^ ± 29.4 ^A’’^	0.124 ^b^ ± 0.007 ^B’’^	0.784 ^a^ ± 0.038 ^A’’^
MS	Control	3.04 ^a^ ± 0.10 ^DE^	0.0384 ^a^ ± 0.0110 ^BC^	1.30 ^a^ ± 0.07 ^A^	89.2 ^a^ ± 4.7 ^A^	445.2 ^a^ ± 18.3 ^A^	105.0 ^a^ ± 9.5 ^B^	0.161 ^a^ ± 0.009 ^B^	0.804 ^a^ ± 0.040 ^A^
	MD	3.93 ^a^ ± 0.79 ^BCD’^	0.0475 ^a^ ± 0.0087 ^ABCD’^	1.63 ^a^ ± 0.11 ^A’^	92.4 ^a^ ± 5.3 ^A’^	452.3 ^a^ ± 30.5 ^A’^	116.5 ^a^ ± 31.1 ^B’^	0.222 ^a^ ± 0.021 ^A’^	0.804 ^a^ ± 0.000 ^A’^
	SD	1.96 ^a^ ± 0.93 ^BC’’^	0.0537 ^a^ ± 0.0103 ^AB’’^	1.05 ^a^ ± 0.07 ^A’’^	45.1 ^a^ ± 2.6 ^A’’^	232.4 ^a^ ± 18.0 ^A’’^	81.8 ^a^ ± 27.6 ^A’’^	0.254 ^a^ ± 0.018 ^B’’^	0.782 ^a^ ± 0.036 ^A’’^
P	Control	12.65 ^a^ ± 3.06 ^AB^	0.0540 ^a^ ± 0.0068 ^A^	0.94 ^a^ ± 0.04 ^A^	127.1 ^a^ ± 6.1 ^A^	809.7 ^a^ ± 39.2 ^A^	132.2 ^a^ ± 54.6 ^B^	0.258 ^a^ ± 0.012 ^B^	0.800 ^a^ ± 0.040 ^A^
	MD	5.06 ^b^ ± 2.95 ^BCD’^	0.0412 ^a^ ± 0.0117 ^D’^	0.59 ^a^ ± 0.06 ^B’^	94.8 ^a^ ± 18.3 ^A’^	620.9 ^a^ ± 153.3 ^A’^	64.6 ^a^ ± 14.5 ^A’^	0.160 ^a^ ± 0.021 ^A’^	0.787 ^a^ ± 0.034 ^A’^
	SD	1.18 ^b^ ± 0.16 ^C’’^	0.0456 ^a^ ± 0.0167 ^CD’’^	0.76 ^a^ ± 0.04 ^A’’^	58.1 ^a^ ± 18.2 ^A’’^	284.7 ^a^ ± 93.3 ^A’’^	46.7 ^a^ ± 3.5 ^A’’^	1.755 ^a^ ± 0.383 ^A’’^	0.623 ^a^ ± 0.023 ^A’’^
S1	Control	8.08 ^a^ ± 1.69 ^CD^	0.0353 ^a^ ± 0.0098 ^CD^	0.67 ^a^ ± 0.04 ^A^	107.5 ^a^ ± 5.5 ^A^	640.2 ^a^ ± 34.6 ^A^	186.1 ^a^ ± 25.6 ^B^	0.213 ^a^ ± 0.010 ^B^	0.835 ^a^ ± 0.042 ^A^
	MD	4.10 ^b^ ± 1.07 ^BCD’^	0.0491 ^a^ ± 0.0090 ^ABCD’^	0.76 ^a^ ± 0.06 ^AB’^	52.8 ^b^ ± 3.1 ^A’^	303.9 ^b^ ± 19.8 ^A’^	35.0 ^a^ ± 18.4 ^B’^	0.191 ^a^ ± 0.009 ^A’^	0.733 ^a^ ± 0.036 ^A’^
	SD	2.20 ^c^ ± 1.03 ^C’’^	0.0466 ^a^ ± 0.0079 ^BC’’^	0.84 ^a^ ± 0.04 ^A’’^	32.3 ^b^ ± 2.0 ^A’’^	173.7 ^b^ ± 13.9 ^A’’^	27.9 ^a^ ± 4.4 ^A’’^	0.329 ^a^ ± 0.024 ^AB’’^	0.724 ^a^ ± 0.034 ^A’’^
S2	Control	12.15 ^a^ ± 1.82 ^ABC^	0.0454 ^a^ ± 0.0034 ^ABC^	0.75 ^a^ ± 0.04 ^A^	157.1 ^a^ ± 8.1 ^A^	879.8 ^a^ ± 39.2 ^A^	253.3 ^a^ ± 52.4 ^B^	0.209 ^a^ ± 0.011 ^B^	0.820 ^a^ ± 0.000 ^A^
	MD	4.18 ^b^ ± 0.70 ^BCD’^	0.0459 ^a^ ± 0.0063 ^BCD’^	0.73 ^a^ ± 0.04 ^B’^	71.2 ^b^ ± 3.6 ^A’^	415.7 ^b^ ± 21.0 ^A’^	83.9 ^b^ ± 19.1 ^B’^	0.226 ^a^ ± 0.010 ^A’^	0.764 ^a^ ± 0.038 ^A’^
	SD	3.54 ^b^ ± 0.90 ^BC’’^	0.0457 ^a^ ± 0.0071 ^BC’’^	0.76 ^a^ ± 0.03 ^A’’^	64.2 ^b^ ± 3.2 ^A’’^	364.2 ^b^ ± 17.8 ^A’’^	54.2 ^b^ ± 16.0 ^A’’^	0.242 ^a^ ± 0.013 ^B’’^	0.778 ^a^ ± 0.038 ^A’’^

**Table 2 plants-12-04136-t002:** Mean values and standard deviation of the parameters obtained from the hyperbolic tangential model fitted to the ETR versus irradiance curves. ETRmax maximum rate of electron transport. Alpha-ETR initial efficiency of photosynthesis in relation to incident light intensity (µmol electrons/µmol photons). ETRd electron transport rate in dark conditions. ISAT_ETR_50_ point at which the electron transport rate reaches 50% of its maximum value in response to light intensity, which, in the case of ISAT_ETR75 and ISAT_ETR99.9, would be 75% and 99.9%, respectively. In each parameter, different lowercase letters indicate significant differences among treatments (“Control”, “MD”, and “SD”) for the same cultivar. Different capital letters indicate significant differences among cultivars for the “Control” treatment, different capital letters with an apostrophe indicate significant differences among cultivars for the “MD” treatment, and different capital letters with two apostrophes indicate significant differences among cultivars for the “SD” treatment.

Cultivar	Condition	ETR_max	Alpha-ETR (α)	ETRd	ISAT_A50	ISAT_A75	ISAT_A99.9
Name	Symbol	µmol e^−^ m^−^²·s^−^¹	µmol e^−^/µmol photons	µmol e^−^ m^−^²·s^−^¹	µmol photons m^−^²·s^−^¹	µmol photons m^−^²·s^−^¹	µmol photons m^−^²·s^−^¹
AE	Control	72.380 ^a^ ± 3.62 ^AB^	0.3400 ^a^ ± 0.0278 ^A^	0.87 ^a^ ± 0.04 ^ABCD^	125.0 ^a^ ± 6.3 ^AB^	219.5 ^a^ ± 11.0 ^AB^	851.8 ^a^ ± 42.6 ^AB^
	MD	42.38 ^b^ ± 1.78 ^AB’^	0.3367 ^a^ ± 0.0270 ^A’^	0.50 ^a^ ± 0.04 ^AB’^	69.3 ^a^ ± 2.9 ^AB’^	121.9 ^a^ ± 5.2 ^AB’^	474.2 ^a^ ± 20.2 ^AB’^
	SD	25.02 ^b^ ± 1.33 ^B’’^	0.2953 ^a^ ± 0.0557 ^A’’^	0.03 ^b^ ± 0.01 ^BC’’^	276.6 ^a^ ± 30.5 ^A’’^	485.1 ^a^ ± 53.3 ^A’’^	1881.4 ^a^ ± 206.4 ^A’’^
Ab	Control	55.03 ^a^ ± 2.75 ^BCD^	0.3324 ^a^ ± 0.0278 ^A^	0.72 ^a^ ± 0.04 ^ABCD^	91.6 ^a^ ± 4.6 ^AB^	161.7 ^a^ ± 8.1 ^AB^	629.7 ^a^ ± 31.5 ^AB^
	MD	32.99 ^b^ ± 1.70 ^B’^	0.3158 ^ab^ ± 0.0184 ^A’^	0.14 ^b^ ± 0.01 ^B’^	57.7 ^ab^ ± 2.8 ^B’^	101.8 ^ab^ ± 5.0 ^B’^	396.5 ^ab^ ± 19.3 ^B’^
	SD	26.88 ^b^ ± 1.29 ^B’’^	0.2947 ^b^ ± 0.0249 ^A’’^	0.07 ^b^ ± 0.01 ^BC’’^	36.3 ^b^ ± 2.3 ^B’’^	64.4 ^b^ ± 3.9 ^B’’^	251.8 ^b^ ± 15.6 ^B’’^
C	Control	69.24 ^a^ ± 3.46 ^ABC^	0.3405 ^a^ ± 0.0090 ^A^	0.96 ^a^ ± 0.05 ^ABC^	113.1 ^a^ ± 5.7 ^AB^	198.8 ^a^ ± 9.9 ^AB^	772.6 ^a^ ± 38.6 ^AB^
	MD	68.21 ^a^ ± 3.75 ^A’^	0.3217 ^a^ ± 0.0227 ^A’^	0.49 ^a^ ± 0.03 ^AB’^	117.8 ^a^ ± 6.4 ^A’^	207.7 ^a^ ± 11.2 ^A’^	808.9 ^a^ ± 43.6 ^A’^
	SD	74.26 ^a^ ± 4.32 ^A’’^	0.3310 ^a^ ± 0.0241 ^A’’^	0.59 ^a^ ± 0.04 ^A’’^	83.8 ^a^ ± 4.8 ^B’’^	147.4 ^a^ ± 8.5 ^B’’^	573.0 ^a^ ± 33.2 ^B’’^
Ch	Control	88.94 ^a^ ± 4.45 ^QB^	0.3340 ^a^ ± 0.0162 ^A^	1.25 ^a^ ± 0.06 ^AB^	133.4 ^a^ ± 6.7 ^AB^	234.5 ^a^ ± 11.7 ^AB^	910.9 ^a^ ± 45.5 ^AB^
	MD	22.96 ^b^ ± 1.11 ^B’^	0.3161 ^a^ ± 0.0298 ^A’^	0.08 ^b^ ± 0.00 ^B’^	45.93 ^b^ ± 2.4 ^B’^	81.0 ^b^ ± 4.1 ^B’^	315.5 ^b^ ± 16.0 ^B’^
	SD	26.17 ^b^ ± 1.51 ^B’’^	0.3051 ^a^ ± 0.0439 ^A’’^	0.24 ^b^ ± 0.02 ^B’’^	47.5 ^b^ ± 2.4 ^B’’^	83.7 ^b^ ± 4.2 ^B’’^	325.5 ^b^ ± 16.4 ^B’’^
CJ	Control	36.10 ^a^ ± 2.12 ^CD^	0.3209 ^a^ ± 0.0238 ^A^	0.34 ^ab^ ± 0.02 ^CDE^	63.7 ^a^ ± 4.5 ^AB^	113.0 ^a^ ± 8.1 ^AB^	441.7 ^a^ ± 31.7 ^AB^
	MD	37.68 ^a^ ± 3.18 ^B’^	0.3452 ^a^ ± 0.0299 ^A’^	0.44 ^a^ ± 0.06 ^AB’^	59.0 ^a^ ± 5.4 ^B’^	103.7 ^a^ ± 9.5 ^B’^	403.2 ^a^ ± 36.8 ^B’^
	SD	19.89 ^a^ ± 1.14 ^BC’’^	0.3010 ^a^ ± 0.0441 ^A’’^	0.03 ^b^ ± 0.00 ^BC’’^	37.3 ^a^ ± 1.8 ^B’’^	65.9 ^a^ ± 3.1 ^B’’^	256.6 ^a^ ± 12.2 ^B’’^
E	Control	94.48 ^a^ ± 4.92 ^A^	0.3479 ^a^ ± 0.0080 ^A^	1.26 ^a^ ± 0.06 ^A^	138.6 ^a^ ± 6.1 ^A^	243.3 ^a^ ± 10.7 ^A^	944.0 ^a^ ± 41.5 ^A^
	MD	39.42 ^b^ ± 1.70 ^B’^	0.3346 ^a^ ± 0.0317 ^A’^	0.38 ^b^ ± 0.03 ^AB’^	37.0 ^b^ ± 3.6 ^B’^	64.7 ^b^ ± 6.5 ^B’^	250.5 ^b^ ± 25.3 ^B’^
	SD	2.09 ^c^ ± 0.11 ^C’’^	0.2960 ^b^ ± 0.0420 ^A’’^	0.17 ^b^ ± 0.01 ^BC’’^	10.5 ^b^ ± 0.9 ^B’’^	18.6 ^b^ ± 1.6 ^B’’^	72.6 ^b^ ± 6.4 ^B’’^
F	Control	31.33 ^a^ ± 1.78 ^CD^	0.3151 ^a^ ± 0.0300 ^A^	0.44 ^ab^ ± 0.02 ^BCDE^	55.3 ^a^ ± 3.1 ^AB^	97.5 ^a^ ± 5.3 ^AB^	379.5 ^a^ ± 20.7 ^AB^
	MD	58.06 ^b^ ± 3.75 ^AB’^	0.3403 ^a^ ± 0.0261 ^A’^	0.66 ^a^ ± 0.05 ^A’^	81.8 ^a^ ± 4.3 ^AB’^	143.8 ^a^ ± 7.5 ^AB’^	558.8 ^a^ ± 29.0 ^AB’^
	SD	31.45 ^a^ ± 1.63 ^B’’^	0.3336 ^a^ ± 0.0291 ^A’’^	0.13 ^b^ ± 0.03 ^BC’’^	35.6 ^a^ ± 2.5 ^B’’^	63.0 ^a^ ± 4.4 ^B’’^	245.7 ^a^ ± 17.2 ^B’’^
H	Control	86.03 ^a^ ± 4.05 ^AB^	0.3276 ^a^ ± 0.0141 ^A^	0.34 ^a^ ± 0.03 ^CDE^	132.3 ^a^ ± 5.9 ^AB^	233.5 ^a^ ± 10.3 ^AB^	909.4 ^a^ ± 40.2 ^AB^
	MD	51.05 ^ab^ ± 4.14 ^AB’^	0.3153 ^a^ ± 0.0301 ^A’^	0.24 ^a^ ± 0.03 ^AB’^	89.0 ^a^ ± 7.9 ^AB’^	157.0 ^a^ ± 14.0 ^AB’^	611.5 ^a^ ± 54.7 ^AB’^
	SD	24.58 ^b^ ± 1.34 ^BC’’^	0.2968 ^a^ ± 0.0345 ^A’’^	0.22 ^a^ ± 0.02 ^BC’’^	69.2 ^a^ ± 6.1 ^B’’^	122.7 ^a^ ± 10.8 ^B’’^	479.1 ^a^ ± 41.8 ^B’’^
K	Control	36.33 ^a^ ± 1.91 ^CD^	0.2925 ^a^ ± 0.0471 ^A^	0.17 ^a^ ± 0.03 ^E^	40.5 ^a^ ± 3.7 ^AB^	71.9 ^a^ ± 6.6 ^AB^	280.9 ^a^ ± 26.0 ^AB^
	MD	44.38 ^a^ ± 2.24 ^AB’^	0.3215 ^a^ ± 0.0283 ^A’^	0.02 ^a^ ± 0.01 ^B’^	62.9 ^a^ ± 2.8 ^AB’^	111.0 ^a^ ± 4.9 ^AB’^	432.6 ^a^ ± 19.4 ^AB’^
	SD	27.02 ^a^ ± 1.76 ^B’’^	0.3248 ^a^ ± 0.0202 ^A’’^	0.05 ^a^ ± 0.01 ^BC’’^	45.6 ^a^ ± 3.1 ^B’’^	80.5 ^a^ ± 5.4 ^B’’^	313.9 ^a^ ± 21.2 ^B’’^
M	Control	52.98 ^a^ ± 2.65 ^BCD^	0.3581 ^a^ ± 0.0277 ^A^	0.98 ^a^ ± 0.04 ^ABC^	115.6 ^a^ ± 7.8 ^AB^	203.1 ^a^ ± 13.7 ^AB^	788.3 ^a^ ± 53.2 ^AB^
	MD	49.11 ^ab^ ± 3.41 ^AB’^	0.3198 ^b^ ± 0.0252 ^A’^	0.30 ^b^ ± 0.04 ^AB’^	94.1 ^a^ ± 6.2 ^AB’^	165.9 ^a^ ± 10.9 ^AB’^	646.0 ^a^ ± 42.5 ^AB’^
	SD	30.13 ^b^ ± 1.68 ^B’’^	0.3029 ^b^ ± 0.0212 ^A’’^	0.02 ^b^ ± 0.00 ^BC’’^	58.9 ^a^ ± 5.0 ^B’’^	104.4 ^a^ ± 8.8 ^B’’^	407.7 ^a^ ± 34.4 ^B’’^
MS	Control	26.06 ^a^ ± 1.22 ^D^	0.3052 ^a^ ± 0.0372 ^A^	0.11 ^ab^ ± 0.01 ^DE^	14.3 ^a^ ± 2.8 ^B^	24.6 ^a^ ± 5.1 ^B^	94.0 ^a^ ± 20.0 ^B^
	MD	40.67 ^b^ ± 2.29 ^AB’^	0.3200 ^a^ ± 0.0155 ^A’^	0.23 ^a^ ± 0.01 ^AB’^	70.5 ^b^ ± 4.2 ^AB’^	124.4 ^b^ ± 7.5 ^AB’^	484.7 ^b^ ± 29.4 ^AB’^
	SD	25.17 ^a^ ± 1.49 ^B’’^	0.3233 ^a^ ± 0.0197 ^A’’^	0.07 ^b^ ± 0.01 ^BC’’^	50.5 ^ab^ ± 3.2 ^B’’^	89.4 ^ab^ ± 5.7 ^B’’^	348.6 ^ab^ ± 22.2 ^B’’^
P	Control	79.79 ^a^ ± 4.10 ^AB^	0.3357 ^a^ ± 0.0146 ^A^	0.99 ^a^ ± 0.06 ^ABC^	123.6 ^a^ ± 5.5 ^AB^	217.9 ^a^ ± 9.7 ^AB^	848.1 ^a^ ± 37.5 ^AB^
	MD	45.69 ^b^ ± 1.92 ^AB’^	0.2925 ^a^ ± 0.0341 ^A’^	0.23 ^b^ ± 0.01 ^AB’^	81.9 ^a^ ± 3.3 ^AB’^	145.6 ^a^ ± 6.8 ^AB’^	569.7 ^a^ ± 26.6 ^AB’^
	SD	8.38 ^c^ ± 0.51 ^C’’^	0.2634 ^b^ ± 0.0372 ^B’’^	0.09 ^c^ ± 0.02 ^C’’^	27.9 ^b^ ± 2.6 ^B’’^	49.7 ^b^ ± 4.7 ^B’’^	195.2 ^b^ ± 18.6 ^B’’^
S1	Control	71.01 ^a^ ± 4.03 ^AB^	0.3560 ^a^ ± 0.0146 ^A^	1.10 ^a^ ± 0.05 ^ABC^	112.1 ^a^ ± 6.6 ^AB^	196.7 ^a^ ± 11.6 ^AB^	762.8 ^a^ ± 45.1 ^AB^
	MD	36.11 ^b^ ± 1.71 ^B’^	0.3358 ^a^ ± 0.0244 ^A’^	0.47 ^b^ ± 0.03 ^AB’^	64.4 ^b^ ± 3.9 ^AB’^	113.3 ^b^ ± 6.8 ^AB’^	440.0 ^b^ ± 26.5 ^AB’^
	SD	29.27 ^b^ ± 1.68 ^B’’^	0.3291 ^a^ ± 0.0237 ^A’’^	0.24 ^b^ ± 0.02 ^B’’^	41.2 ^b^ ± 2.4 ^B’’^	72.7 ^b^ ± 4.3 ^B’’^	283.2 ^b^ ± 16.8 ^B’’^
S2	Control	69.06 ^a^ ± 3.34 ^ABC^	0.3443 ^a^ ± 0.0056 ^A^	0.69 ^a^ ± 0.04 ^ABCD^	111.2 ^a^ ± 5.3 ^AB^	195.8 ^a^ ± 9.3 ^AB^	761.3 ^a^ ± 36.0 ^AB^
	MD	28.51 ^b^ ± 1.59 ^B’^	0.3183 ^b^ ± 0.0129 ^A’^	0.22 ^b^ ± 0.01 ^AB’^	49.7 ^b^ ± 2.5 ^B’^	87.6 ^b^ ± 4.4 ^B’^	341.0 ^b^ ± 17.0 ^B’^
	SD	29.73 ^b^ ± 1.61 ^B’’^	0.3284 ^ab^ ± 0.01659 ^A’’^	0.17 ^b^ ± 0.01 ^BC’’^	50.0 ^b^ ± 3.1 ^B’’^	88.2 ^b^ ± 5.4 ^B’’^	343.7 ^b^ ± 21.3 ^B’’^

**Table 3 plants-12-04136-t003:** Mean values and standard deviation of the parameters obtained from the hyperbolic tangential model fitted to the ETR versus irradiance curves. The number of photons required to move an electron is calculated as the inverse of the alpha_ETR parameter (µmol photons/µmol e^−^). The number of photons to fix a molecule of CO_2_ is calculated from the inverse of the φ_PSII parameter (µmol photons/µmol CO_2)_. The number of electrons required to fix a CO_2_ molecule (mol e^−^/mol CO_2_) is calculated by dividing alpha_ETR by φ_PSII (µmol e^−^/µmol CO_2_). Fv/Fm refers to the ratio between the maximum variability of fluorescence (Fm) and the steady-state fluorescence (F0). ΔF/Fm′ maximum quantum efficiency of photosystem II (PSII) in photosynthesis (effective quantum yield). NPQ (Non-Photochemical Quenching) is the capacity of PSII to dissipate excess energy as heat. qP (Photochemical Quenching) represents the fraction of PSII reaction centres that are in the reduced state, i.e., available for the photochemical process, and qN (Non-Photochemical Quenching) represents the fraction of reaction centres of PSII that are in an oxidised state. In each parameter, different lowercase letters indicate significant differences among treatments (“Control”, “MD”, and “SD”) for the same cultivar. Different capital letters indicate significant differences among cultivars for the “Control” treatment, different capital letters with an apostrophe indicate significant differences among cultivars for the “MD” treatment, and different capital letters with two apostrophes indicate significant differences among cultivars for the “SD” treatment.

Cultivar	Condition	Fv/Fm	ΔF/Fm′	NPQ	qP	qN	mol photons/mol e^−^	mol photons/mol CO₂	mol e^−^/mol CO₂
Name	Symbol	-	-	-	-	-	µmol photons/µmol e^−^	µmol photons/µmol CO₂	µmol e^−^/µmol CO₂
AE	Control	0.827 ^a^ ± 0.041 ^AB^	0.583 ^a^ ± 0.029 ^AB^	2.47 ^a^ ± 0.12 ^AB^	0.388 ^a^ ± 0.019 ^AB^	0.803 ^a^ ± 0.04 ^A^	2.94 ^a^ ± 0.03 ^B^	20.18 ^a^ ± 1.49 ^A^	6.86 ^a^ ± 0.04 ^A^
	MD	0.830 ^a^ ± 0.041 ^A’^	0.599 ^a^ ± 0.026 ^ABC’^	2.34 ^a^ ± 0.14 ^ABC’^	0.366 ^a^ ± 0.019 ^BCDE’^	0.788 ^a^ ± 0.038 ^A’^	2.97 ^a^ ± 0.03 ^B’^	23.51 ^a^ ± 7.37 ^CD’^	7.92 ^a^ ± 0.20 ^B’^
	SD	0.832 ^a^ ± 0.000 ^A’’^	0.561 ^a^ ± 0.029 ^AB’’^	2.86 ^a^ ± 0.13 ^ABC’’^	0.346 ^a^ ± 0.023 ^AB’’^	0.830 ^a^ ± 0.040 ^A’’^	3.39 ^a^ ± 0.06 ^B’’^	30.66 ^a^ ± 8.17 ^BC’’^	9.06 ^a^ ± 0.46 ^B’’^
Ab	Control	0.829 ^a^ ± 0.041 ^AB^	0.552 ^a^ ± 0.028 ^ABC^	3.10 ^a^ ± 0.15 ^A^	0.400 ^a^ ± 0.020 ^AB^	0.841 ^a^ ± 0.042 ^A^	3.01 ^a^ ± 0.03 ^B^	31.41 ^a^ ± 6.71 ^A^	10.44 ^a^ ± 0.19 ^A^
	MD	0.769 ^ab^ ± 0.038 ^ABC’^	0.533 ^a^ ± 0.028 ^BCD’^	2.33 ^b^ ± 0.14 ^ABC’^	0.412 ^a^ ± 0.021 ^ABCD’^	0.777 ^a^ ± 0.038 ^A’^	3.17 ^ab^ ± 0.02 ^AB’^	23.87 ^a^ ± 7.38 ^CD’^	7.54 ^ab^ ± 0.14 ^AB’^
	SD	0.742 ^b^ ± 0.036 ^AB’’^	0.505 ^a^ ± 0.023 ^AB’’^	1.83 ^b^ ± 0.11 ^DE’’^	0.434 ^a^ ± 0.025 ^A’’^	0.774 ^a^ ± 0.037 ^AB’’^	3.39 ^b^ ± 0.03 ^B’’^	22.78 ^a^ ± 3.59 ^BC’’^	6.71 ^b^ ± 0.09 ^B’’^
C	Control	0.836 ^a^ ± 0.042 ^A^	0.631 ^a^ ± 0.032 ^A^	2.42 ^a^ ± 0.12 ^AB^	0.352 ^a^ ± 0.018 ^AB^	0.790 ^a^ ± 0.040 ^A^	2.94 ^a^ ± 0.01 ^B^	23.193 ^a^ ± 3.3 ^A^	7.90 ^a^ ± 0.03 ^A^
	MD	0.804 ^a^ ± 0.039 ^ABC’^	0.603 ^a^ ± 0.032 ^AB’^	1.60 ^a^ ± 0.09 ^CD’^	0.313 ^a^ ± 0.017 ^DE’^	0.716 ^a^ ± 0.036 ^A’^	3.11 ^a^ ± 0.02 ^AB’^	20.42 ^a^ ± 4.61 ^CD’^	6.57 ^a^ ± 0.11 ^AB’^
	SD	0.809 ^a^ ± 0.039 ^AB’’^	0.576 ^a^ ± 0.030 ^A’’^	2.06 ^a^ ± 0.11 ^CDE’’^	0.373 ^a^ ± 0.023 ^A’’^	0.779 ^a^ ± 0.038 ^AB’’^	3.02 ^a^ ± 0.02 ^B’’^	23.80 ^a^ ± 3.52 ^B’’^	7.88 ^a^ ± 0.09 ^B’’^
Ch	Control	0.804 ^a^ ± 0.040 ^AB^	0.549 ^a^ ± 0.027 ^ABC^	2.38 ^a^ ± 0.12 ^AB^	0.336 ^a^ ± 0.017 ^B^	0.812 ^a^ ± 0.041 ^A^	2.99 ^a^ ± 0.02 ^B^	18.10 ^a^ ± 1.46 ^A^	6.05 ^a^ ± 0.02 ^A^
	MD	0.715 ^a^ ± 0.037 ^C’^	0.536 ^a^ ± 0.027 ^BCD’^	1.90 ^ab^ ± 0.08 ^BCD’^	0.406 ^a^ ± 0.022 ^ABCDE’^	0.753 ^a^ ± 0.038 ^A’^	3.16 ^a^ ± 0.03 ^AB’^	31.02 ^a^ ± 6.20 ^A’^	9.81 ^a^ ± 0.19 ^AB’^
	SD	0.709 ^a^ ± 0.034 ^AB’’^	0.519 ^a^ ± 0.025 ^AB’’^	1.16 ^b^ ± 0.06 ^E’’^	0.414 ^a^ ± 0.030 ^A’’^	0.681 ^a^ ± 0.030 ^B’’^	3.28 ^a^ ± 0.04 ^B’’^	34.72 ^a^ ± 6.22 ^A’’^	10.59 ^a^ ± 0.27 ^B’’^
CJ	Control	0.824 ^a^ ± 0.000 ^AB^	0.551 ^a^ ± 0.027 ^ABC^	2.80 ^a^ ± 0.16 ^AB^	0.392 ^a^ ± 0.020 ^AB^	0.831 ^a^ ± 0.042 ^A^	3.12 ^a^ ± 0.02 ^B^	25.64 ^a^ ± 3.35 ^A^	8.23 ^a^ ± 0.08 ^A^
	MD	0.808 ^ab^ ± 0.038 ^ABC’^	0.556 ^a^ ± 0.025 ^BCD’^	2.39 ^ab^ ± 0.11 ^ABC’^	0.516 ^b^ ± 0.027 ^A’^	0.808 ^a^ ± 0.040 ^A’^	2.90 ^ab^ ± 0.03 ^B’^	22.27 ^a^ ± 3.25 ^ABC’^	7.69 ^ab^ ± 0.10 ^B’^
	SD	0.770 ^b^ ± 0.039 ^AB’’^	0.506 ^a^ ± 0.026 ^AB’’^	2.03 ^b^ ± 0.10 ^CD E’’^	0.361 ^a^ ± 0.015 ^AB’’^	0.809 ^a^ ± 0.040 ^AB’’^	3.33 ^b^ ± 0.04 ^B’’^	22.13 ^a^ ± 1.46 ^BCD’’^	6.65 ^b^ ± 0.07 ^B’’^
E	Control	0.826 ^a^ ± 0.000 ^AB^	0.565 ^a^ ± 0.028 ^ABC^	2.51 ^a^ ± 0.13 ^AB^	0.393 ^a^ ± 0.020 ^AB^	0.821 ^a^ ± 0.041 ^A^	2.88 ^a^ ± 0.01 ^B^	19.86 ^a^ ± 2.43 ^A^	6.91 ^a^ ± 0.02 ^A^
	MD	0.814 ^a^ ± 0.037 ^AB’^	0.539 ^a^ ± 0.028 ^BCD’^	3.05 ^ab^ ± 0.15 ^A’^	0.464 ^a^ ± 0.025 ^ABC’^	0.846 ^ab^ ± 0.042 ^A’^	3.00 ^a^ ± 0.03 ^B’^	16.52 ^a^ ± 1.79 ^D’^	5.51 ^a^ ± 0.06 ^B’^
	SD	0.462 ^a^ ± 0.029 ^AB’’^	0.516 ^a^ ± 0.026 ^AB’’^	3.44 ^b^ ± 0.17 ^A’’^	0.450 ^a^ ± 0.023 ^A’’^	0.869 ^b^ ± 0.043 ^A’’^	3.38 ^a^ ± 0.04 ^B’’^	14.78 ^a^ ± 1.52 ^D’’^	4.37 ^a^ ± 0.07 ^B’’^
F	Control	0.820 ^a^ ± 0.000 ^AB^	0.539 ^a^ ± 0.027 ^ABC^	2.71 ^a^ ± 0.13 ^AB^	0.451 ^a^ ± 0.025 ^AB^	0.840 ^a^ ± 0.042 ^A^	3.17 ^a^ ± 0.03 ^B^	21.58 ^a^ ± 3.28 ^A^	6.80 ^a^ ± 0.10 ^A^
	MD	0.814 ^a^ ± 0.039 ^AB’^	0.586 ^a^ ± 0.030 ^ABCD’^	2.19 ^b^ ± 0.11 ^BC’^	0.374 ^a^ ± 0.021 ^CDE’^	0.784 ^a^ ± 0.039 ^A’^	2.94 ^a^ ± 0.03 ^B’^	19.04 ^a^ ± 3.46 ^CD’^	6.48 ^a^ ± 0.09 ^B’^
	SD	0.803 ^a^ ± 0.038 ^AB’’^	0.574 ^a^ ± 0.028 ^AB’’’^	2.23 ^b^ ± 0.10 ^BCD’’^	0.438 ^a^ ± 0.029 ^A’’^	0.783 ^a^ ± 0.037 ^AB’’^	3.00 ^a^ ± 0.03 ^B’’^	16.11 ^a^ ± 2.17 ^D’’^	5.37 ^a^ ± 0.06 ^B’’^
H	Control	0.825 ^a^ ± 0.000 ^AB^	0.562 ^a^ ± 0.029 ^ABC^	2.88 ^a^ ± 0.14 ^AB^	0.438 ^a^ ± 0.022 ^AB^	0.819 ^a^ ± 0.040 ^A^	3.05 ^a^ ± 0.01 ^B^	28.53 ^a^ ± 3.22 ^A^	9.35 ^a^ ± 0.05 ^A^
	MD	0.825 ^a^ ± 0.000 ^AB’^	0.580 ^a^ ± 0.031 ^ABCD’^	2.39 ^a^ ± 0.10 ^ABC’^	0.433 ^a^ ± 0.022 ^ABCD’^	0.797 ^a^ ± 0.037 ^A’^	3.17 ^a^ ± 0.03 ^AB’^	20.93 ^a^ ± 5.70 ^ABCD’^	6.60 ^a^ ± 0.17 ^AB’^
	SD	0.825 ^a^ ± 0.000 ^AB’’^	0.567 ^a^ ± 0.029 ^AB’’^	2.58 ^a^ ± 0.13 ^ABCD’’^	0.397 ^a^ ± 0.022 ^A’’^	0.818 ^a^ ± 0.041 ^AB’’^	3.37 ^a^ ± 0.03 ^B’’^	35.98 ^a^ ± 10.75 ^AB’’^	10.68 ^a^ ± 0.37 ^B’’^
K	Control	0.798 ^a^ ± 0.039 ^B^	0.487 ^a^ ± 0.024 ^C^	2.98 ^a^ ± 0.18 ^A^	0.411 ^a^ ± 0.015 ^AB^	0.865 ^a^ ± 0.044 ^A^	3.42 ^a^ ± 0.05 ^A^	65.16 ^a^ ± 24.13 ^A^	19.06 ^a^ ± 1.14 ^A^
	MD	0.731 ^a^ ± 0.036 ^BC’^	0.480 ^a^ ± 0.024 ^D’^	2.10 ^a^ ± 0.11 ^BC’^	0.452 ^a^ ± 0.025 ^ABCD’^	0.806 ^a^ ± 0.040 ^A’^	3.11 ^a^ ± 0.03 ^AB’^	23.92 ^b^ ± 8.55 ^CD’^	7.69 ^a^ ± 0.24 ^AB’^
	SD	0.713 ^a^ ± 0.037 ^AB’’^	0.467 ^a^ ± 0.024 ^B’’^	1.94 ^a^ ± 0.11 ^CDE’’^	0.436 ^a^ ± 0.021 ^A’’^	0.793 ^a^ ± 0.039 ^AB’’^	3.08 ^a^ ± 0.02 ^B’’^	21.80 ^b^ ± 6.17 ^BC’’^	7.08 ^a^ ± 0.12 ^B’’^
M	Control	0.803 ^a^ ± 0.040 ^AB^	0.569 ^a^ ± 0.028 ^ABC^	2.11 ^a^ ± 0.10 ^BC^	0.474 ^a^ ± 0.024 ^A^	0.787 ^a^ ± 0.039 ^AB^	2.79 ^a^ ± 0.03 ^B^	21.57 ^a^ ± 2.08 ^A^	7.73 ^a^ ± 0.06 ^A^
	MD	0.770 ^a^ ± 0.039 ^ABC’^	0.646 ^a^ ± 0.033 ^A’^	0.76 ^b^ ± 0.03 ^D’^	0.344 ^a^ ± 0.024 ^CDE’^	0.457 ^b^ ± 0.012 ^B’^	3.13 ^ab^ ± 0.03 ^AB’^	22.39 ^a^ ± 3.94 ^BCD’^	7.15 ^ab^ ± 0.10 ^AB’^
	SD	0.790 ^a^ ± 0.057 ^AB’’^	0.596 ^a^ ± 0.037 ^A’’^	1.97 ^a^ ± 0.12 ^CDE’’^	0.379 ^a^ ± 0.027 ^A’’^	0.718 ^ab^ ± 0.048 ^AB’’^	3.30 ^b^ ± 0.021 ^B’’^	27.43 ^a^ ± 7.96 ^B’’^	8.31 ^b^ ± 0.17 ^B’’^
MS	Control	0.792 ^a^ ± 0.039 ^B^	0.515 ^a^ ± 0.026 ^BC^	2.87 ^a^ ± 0.10 ^AB^	0.363 ^a^ ± 0.018 ^AB^	0.848 ^a^ ± 0.042 ^A^	3.28 ^a^ ± 0.04 ^B^	28.01 ^a^ ± 9.30 ^A^	8.55 ^a^ ± 0.34 ^A^
	MD	0.804 ^a^ ± 0.000 ^ABC’^	0.571 ^b^ ± 0.027 ^ABCD’^	2.07 ^b^ ± 0.12 ^BC’^	0.396 ^a^ ± 0.023 ^BCDE’^	0.780 ^b^ ± 0.041 ^A’^	3.13 ^a^ ± 0.02 ^AB’^	21.54 ^ab^ ± 3.49 ^BCD’^	6.89 ^a^ ± 0.05 ^AB’^
	SD	0.804 ^a^ ± 0.000 ^AB’’^	0.580 ^b^ ± 0.029 ^A’’^	1.96 ^b^ ± 0.10 ^CDE’’^	0.409 ^a^ ± 0.025 ^A’’^	0.770 ^b^ ± 0.039 ^AB’’^	3.09 ^a^ ± 0.02 ^B’’^	19.30 ^b^ ± 4.13 ^CD’’^	6.24 ^a^ ± 0.08 ^B’’^
P	Control	0.800 ^a^ ± 0.040 ^B^	0.612 ^a^ ± 0.031 ^A^	1.44 ^a^ ± 0.07 ^C^	0.374 ^a^ ± 0.020 ^AB^	0.711 ^a^ ± 0.037 ^B^	2.98 ^a^ ± 0.02 ^B^	18.80 ^a^ ± 2.64 ^A^	6.31 ^a^ ± 0.04 ^A^
	MD	0.819 ^a^ ± 0.000 ^AB’^	0.593 ^a^ ± 0.029 ^ABC’^	2.11 ^a^ ± 0.13 ^BC’^	0.263 ^ab^ ± 0.009 E^’^	0.759 ^ab^ ± 0.039 ^A’^	3.42 ^a^ ± 0.03 ^A’^	25.46 ^a^ ± 6.09 ^AB’^	7.45 ^a^ ± 0.21 ^A’^
	SD	0.601 ^a^ ± 0.020 ^B’’^	0.554 ^a^ ± 0.029 ^AB’’^	3.01 ^b^ ± 0.15 ^AB’’^	0.213 ^b^ ± 0.008 ^B’’^	0.858 ^b^ ± 0.046 ^A’’^	3.80 ^a^ ± 0.03 ^A’’^	27.52 ^a^ ± 8.88 ^AB’’^	7.24 ^a^ ± 0.26 ^A’’^
S1	Control	0.835 ^a^ ± 0.042 ^A^	0.525 ^a^ ± 0.024 ^BC^	3.16 ^a^ ± 0.16 ^A^	0.389 ^a^ ± 0.019 ^AB^	0.866 ^a^ ± 0.045 ^A^	2.81 ^a^ ± 0.02 ^B^	29.68 ^a^ ± 7.52 ^A^	10.57 ^a^ ± 0.11 ^A^
	MD	0.772 ^a^ ± 0.035 ^ABC’^	0.494 ^a^ ± 0.027 ^D’^	2.79 ^a^ ± 0.20 ^AB’^	0.441 ^a^ ± 0.026 ^ABCD’^	0.836 ^a^ ± 0.043 ^A’^	2.98 ^a^ ± 0.02 ^B’^	21.04 ^ab^ ± 3.59 ^BCD’^	7.06 ^a^ ± 0.09 ^B’^
	SD	0.794 ^a^ ± 0.040 ^AB’’^	0.521 ^a^ ± 0.026 ^AB’’^	2.59 ^a^ ± 0.13 ^ABCD’’^	0.478 ^a^ ± 0.024 ^A’’^	0.825 ^a^ ± 0.041 ^A’’^	3.04 ^a^ ± 0.02 ^B’’^	21.97 ^b^ ± 3.75 ^BCD’’^	7.24 ^a^ ± 0.09 ^B’’^
S2	Control	0.820 ^a^ ± 0.000 ^AB^	0.549 ^a^ ± 0.028 ^ABC^	2.74 ^a^ ± 0.14 ^AB^	0.411 ^a^ ± 0.021 ^AB^	0.833 ^a^ ± 0.041 ^A^	2.90 ^a^ ± 0.01 ^B^	22.12 ^a^ ± 1.62 ^A^	7.62 ^a^ ± 0.01 ^A^
	MD	0.764 ^a^ ± 0.038 ^ABC’^	0.513 ^a^ ± 0.027 ^CD’^	2.05 ^a^ ± 0.12 ^BC’^	0.492 ^a^ ± 0.019 ^AB’^	0.817 ^a^ ± 0.045 ^A’^	3.14 ^b^ ± 0.01 ^AB’^	22.13 ^a^ ± 2.81 ^BCD’^	7.043 ^b^ ± 0.04 ^AB’^
	SD	0.778 ^a^ ± 0.038 ^AB’’^	0.522 ^a^ ± 0.025 ^AB’’^	2.32 ^a^ ± 0.14 ^ABCD’’^	0.470 ^a^ ± 0.0230 ^A’’^	0.808 ^a^ ± 0.041 ^AB’’^	3.05 ^ab^ ± 0.02 ^B’’^	22.34 ^a^ ± 3.41 ^BCD’’^	7.34 ^ab^ ± 0.05 ^B’’^

**Table 4 plants-12-04136-t004:** Classification of the studied cultivars according to their tolerance to the deficit of water and their optimal light intensity interval of growth.

	Arid Areas 200–400 mm/Year	Moderately Tolerant 350–500 mm/Year	Moderately Sensitive 400–650 mm/Year	Non-Arid Areas 500–800 mm/Year
Low irradiances	-	F	MS	CJ
Medium irradiances	AO	S1		K
High irradiances	M, S2, C	CH, P	AE, H	E

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
