# Peer review of "Study of the Photosynthesis Response during the Gradual Lack of Water for 14 Olea europaea L. subsp europaea Cultivars and Their Adaptation to Climate Change"

_plants, 2023, doi:10.3390/plants12244136_

Round 1

Reviewer 1 Report

Comments and Suggestions for Authors

In the present manuscript, authors have shown “Study of photosynthesis-light response curves in Olea europaea L. subsp europaea cultivars and their adaptation to climate change”.

Actually, there are many flaws in this MS that it is not suitable for publication in Plants in its current form.

The main concern is:

Abstract:

(1) The abstract is very poorly constructed. For example, the basic experiment that was done to conclude their conclusion and the obtained results are not clearly revealed in the abstract. This part needs to be completely re-written.

Introduction:

(2) The introduction is written chaotically. There is a lack of information about what is new. Please incorporate recent reference.

Results:

(3) Result section is weakly presented without any quantitative data.
(4)
In Tables: The presentation of the statistical analysis is incomplete. The authors should display the differences with different letters.

Discussion:

(5) This manuscript lacks discussion and reference of the results obtained to previously published studies. There is not even a discussion chapter. This situation is unacceptable for a research article. I ask the authors to add this chapter and to relate their research findings to world publications. Authors should discuss how their results fit into the large pools of study.

Material and methods:

(6) In material and methods, please cite all described methods because these methods are not your own.

Conclusion:

(7) The conclusion section is poorly written. Please list the main conclusions of the research.

References:

(8) References need to be cross-checked.

Linguistic quality:

 (9) The language quality is so poor and this paper must be edited by professional English editor.

Comments on the Quality of English Language

Extensive editing of English language required

Author Response

The comments to your suggestions are in the file attached

Reviewer 2 Report

Comments and Suggestions for Authors

General comments

I have read the manuscript:  Plants MDPI. Entitle: Study of photosynthesis-light response curves in Olea europaea L. sub. sp europaea cultivars and their adaptation to climate change written by Genoveva Carmen Martos de la Fuente et. al., for publication of Plant MDPI. In this study, author investigated the photosynthesis response curves using different light intensities in 14 olive cultivars subjected to moderate and severe water deficits.

The overall research is well conducted but author should further improvement of this manuscript for journal acceptance. Research is obvious application potential for the readers because this research provides the important finding of Olive cultivars under the different intensities of drought stress condition and allow the tolerance to water stress. In this sense, this manuscript is much more valuable. However, I found a lack of story connection and lack of potential references (some I suggested some below). Overall after I evaluate and request the author for this manuscript as a “MAJOR REVISION”.

Major Suggestions

1) Abstract: Author should improve the abstract further. Abstract should be more informative, include the clear result and common message for audiences. This is very important than only lengthening the text. Author should work in novelty; how does the study elucidate your finding useful for the society by reducing the text of methodology part.

2) Hypothesis of the Review: Author not clearly mentions research hypothesis of this paper. Please mention clear hypothesis and connect this with the research hypothesis in the last section of the introduction section. The hypothesis should be very clear because, without appropriate literature, questions, or hypotheses in the introduction section the entire text will be unclear.

3) Concise the text: author should concise the text by removing the unnecessary and less important text. Please include the text related on the research title and its circumstances by cutback unnecessary text.

 Some other line to line comments

4) Line no. 23 (Introduction): Generally, under limited water (water scarcity) leaves are primarily affected which is pivotal role in plant. Refer DOI:10.1016/j.scienta.2018.11.021 “under drought stress condition gs showed the primary responses by closing of stomata and its showed directly effect on the reduction of the photosynthesis” Author should be include this information which better clarify the plant-water relation.

5) Line no. 68-71 : author should further improve the Line no. 68-71 include more references. The leaf anatomy specially its internal structure is very important for the photosynthesis efficiency. Refer this article as a reference. DOI:10.1016/j.scienta.2018.11.021 “Increase the palisade mesophyll parenchyma enhance the photosynthesis rate because its help to better capture the light and easily assist to change the light energy to chemical energy while performing the photosynthesis”.  

6) Discussion: Author should significant improve the discussion section while they are describe the photosynthesis light response curve by referring the articles which presented the data related to the “photosynthesis light reference curve” Refer articles 1) https://doi.org/10.1016/j.foreco.2020.118099

7) Line no. 386 (Results): Table 3, legend is too long by making it concise form but please do not loose related full information what presented in the table. Accordingly please check all the table and Figure legends.

8) Line no. 487- 488 (MM section): please correct the manuscript carefully, all the notation and formula related equation “Pmax (also known as Psat or PN, mol CO2 m-2 s-1)” it is not correct, please also correct the subscript and superscript as well.

9) Line no. 570 (MM section): Please concise the text by removing the unnecessary text in “statistical analysis” also do not change the paragraphs more. “Once all the study parameters, both those from non-linear fitting models and those from the analysis with the LI-COR 6800 system, were collected” it is not need in this section, please cut and make more concise further.

10) Line no. 601 (Conclusion): I did not see the conclusion section separately made by author. Please remember that conclusion should not be repetitive in the abstract or a summary of the results section. I would love to read striking points and take-home messages that will linger in the readers’ minds. What is the novelty, how does the study elucidate some questions in this field, and the contributions the paper may offer to the scientific community?

11) Line no. 630 (References): please include more related citation, check their pattern and writing style, spell check, and other grammatical errors. moreover, the author should cut the old and less matching literature and include the latest literature some of them are above.

Good Luck !

Author Response

Te reply to your suggestions is in the file attached, dear reviewer

Round 2

Reviewer 1 Report

Comments and Suggestions for Authors

The comments have been mostly addressed. The manuscript has improved and can be published.

Comments on the Quality of English Language

Minor editing of English language required

Author Response

Thank you very much dear reviewer. Your contributions have been very valuable to us

Reviewer 2 Report

Comments and Suggestions for Authors

Dear Author

I have read the revised manuscript plants-2667778. Entitled: Study of photosynthesis-light response curves in Olea europaea L. sub sp europaea cultivars and their adaptation to climate change in plant MDPI. This is the second submission made by the author. The author addressed all the questions and suggestions that I raised the issue in the review of the original manuscript but adress the comment 6 is still not enough regarding to address the light response curve. "Refer to article  https://doi.org/10.1016/j.foreco.2020.118099, mention that increase the Pn capacity with increase the PPFDs levels". Apart from above comments, author improved the abstract significantly. Author significantly improved their research hypothesis and well connected with the research objectives in this time. This manuscript improved the flow of writing, which was comparatively shallow in the original version but in this revised copy author very well addressed all the quarries and suggestions. Before accepting this manuscript, please check again the referencing. Further if there is anything needed to be revised by the author, especially English grammar, or spell check, I request this manuscript is currently in “Minor Revision” and the author may correct any further grammatical errors (if any) the author may improve in this stage.

Thank you.

Author Response

Thank you very much, dear reviewer. Your contributions have been very valuable to us. We have revised the English in the paragraphs in red and have corrected the sentences in blue as can be seen in the text. We have also considered the phrase that you tell us about the increase in photosynthesis with irradiance (lines 274 and 275)